# Alzheimer's Aβ catalyzes Tau phase separation and aggregation via early nanocluster solubilization

Xun Sun[1,6], Yiming Tang [2,3,6], Xue Wang[1,6], Guadalupe Pereira Curia[1], Rebecca Sternke-Hoffmann [1], Cecilia Mörman[1,4], Juan Atilio Gerez [5], Roland Riek [5], Guanghong Wei [2,3] ✉ & Jinghui Luo [1] ✉

Extracellular amyloid-beta (Aβ) plaques and intracellular neurofibrillary tangles (NFTs) composed of hyperphosphorylated Tau are the two main pathological hallmarks of Alzheimer's disease (AD). Although the co-occurrence and synergistic effects of Aβ and Tau are well established, the mechanisms underlying their interplay in a biomolecular condensate environment remain unclear. Here we show that Aβ40 does not undergo liquid–liquid phase separation (LLPS) but significantly enhances Tau phase separation and is recruited into Tau condensates. This recruitment alters condensate physicochemical properties, accelerates liquid-to-solid maturation, promotes Tau amyloid fibril formation, and increases Tau-mediated cytotoxicity. Notably, prior to condensate formation, Aβ40 transiently solubilizes Tau nanoclusters into smaller species. Simulations further indicate that early interactions are non-specific and mediated by Tau repeat domains, ultimately promoting pathogenic aggregation. These findings support a model wherein Aβ act as a catalyst for Tau condensation and fibrillation towards pathological aggregates by solubilizing Tau nanoclusters during early phase interactions.

Alzheimer's disease (AD) is a progressive neurodegenerative disorder characterized by cognitive decline, with its pathology marked by extracellular amyloid-beta (Aβ) plaques and intracellular neurofibrillary tangles (NFTs) composed of Tau protein[1]. Aβ peptides, ranging from 39 to 43 residues, are generated through the enzymatic cleavage of amyloid precursor protein (APP) by β- and γ-secretases[2]. Aβ40, the predominant form in cerebrospinal fluid, differs from Aβ42 by two C-terminal residues, which enhance Aβ42's aggregation propensity and neurotoxicity[3,4]. Aβ peptides are intrinsically disordered, negatively charged at physiological pH, and prone to forming amyloid fibrils. Notably, Aβ plaque quantity does not correlate directly with cognitive decline severity in AD[5]. Conversely, NFTs strongly correlate with disease progression and cognitive impairment[5]. NFTs, primarily composed of Tau proteins from the microtubule-associated protein (MAP) family, impair microtubule stability and disrupt neuronal function. Alternative splicing of Tau mRNA gives rise to six distinct isoforms[6]. The full-length Tau441 (441 residues, positively charged at physiological pH) is the most prevalent in AD and K18, a truncated Tau model variant containing only the microtubule-binding domain, readily forms fibrils[7].

Although traditionally examined as separate pathological entities, growing evidence indicates that Aβ and Tau exhibit spatial and biochemical interdependence, interacting in ways that influence one other's pathological roles in the brain[8,9]. While Aβ accumulates diffusely across the neocortex, tau pathology typically initiates focally in the medial temporal lobe and then spreads transneuronally to

[1]Center for Life Sciences, Paul Scherrer Institute, Villigen PSI, Switzerland. [2]Key Laboratory for Computational Physical Sciences of Ministry of Education, Fudan University, Shanghai, PR China. [3]State Key Laboratory of Surface Physics, Department of Physics, Fudan University, Shanghai, PR China. [4]Department of Medicine Huddinge, Karolinska Institutet, Huddinge, Sweden. [5]Institute of Molecular Physical Science, Department of Chemistry and Applied Biosciences, ETH Zurich, Zurich, Switzerland. [6]These authors contributed equally: Xun Sun, Yiming Tang, Xue Wang. ✉e-mail: ghwei@fudan.edu.cn; jinghui.luo@psi.ch

connected brain regions[9]. This spread is facilitated by Aβ-induced neuronal hyperactivity and hyperconnectivity, which acts as a bridge between these two hallmark pathologies[10]. The co-localization of Aβ and Tau could facilitate the cross-seeding of pathologies where each protein influences the aggregation or misfolding of the other. Aβ induces Tau hyperphosphorylation, a critical step in NFT formation[11], while Tau modulates Aβ oligomer formation and fibrillation at the oligomeric stage in vitro[12]. Moreover, Tau is essential for Aβ-induced neurotoxicity[13], implying an active, bidirectional pathological interaction rather than a linear cascade. Despite these biochemical and pathological observations, the precise mechanisms by which Aβ influences Tau phase separation and its transition toward aggregated, disease-associated states remain poorly understood.

Liquid-liquid phase separation (LLPS), a biophysical process increasingly implicated in neurodegenerative diseases, occurs when biomolecules such as intrinsically disordered proteins (IDPs) phase separate from bulk solution to form dense liquid droplets[14]. While LLPS is crucial for cellular membrane-less compartmentalization, its dysregulation may contribute to disease pathogenesis[15]. Emerging evidence suggests that LLPS can be preceded by the formation of nanoscale, liquid-like clusters termed nanoclusters[16]. These transient assemblies exist in dynamic equilibrium with their monomeric or oligomeric forms, but some can overcome energetic barriers to form more microscale structures, potentially serving as intermediates in the initiation of phase separation[16]. There is ongoing debate whether Aβ undergoes LLPS as an intermediate step prior to amyloid plaque formation. It also remains unclear whether molecular crowding agents modulate local Aβ concentrations in a manner that accelerates its fibrillization[17,18]. Similarly, Tau's LLPS propensity, influenced by factors such as phosphorylation and ionic strength, may facilitate NFT formation by concentrating Tau in subcellular regions[19,20]. Experimental evidence suggests that aberrant Tau LLPS may precede aggregation[20,21], as some protein droplets transition into insoluble fibrils[22–25]. Phosphorylation or disease-related Tau induces condensation within cells occurs even in the normal physiological environment with the estimated intraneuronal concentration, suggesting that LLPS could be a natural mechanism for organizing Tau within neurons[20]. However, LLPS and amyloid aggregation have also been reported as independent processes[26], and a definitive causal link between LLPS and amyloid formation remains elusive. Additionally, effective strategies for regulating these processes are yet to be established. The interplay between Aβ and Tau in LLPS is particularly intriguing, as their interactions may modulate the pathological deposition, co-aggregation, or toxicity of each protein. Notably, co-localization or heterotypic assemblies of Aβ and Tau has been observed in the brains of Alzheimer's disease patients, suggesting a cooperative role in disease progression[10,20]. Elucidating the molecular basis of this crosstalk in the environment of cellular phase or cellular-like phase is therefore critical for understanding AD pathogenesis and identifying therapeutic targets that disrupt early co-condensation and co-aggregation events.

In this study, we investigated how Aβ40 influences Tau phase separation, phase transition, aggregation, and associated cytotoxicity by employing a combination of microscopy, biochemical-, and biophysical approaches. Our data from mass photometry and NMR suggest that Aβ40 initially facilitates the solubilization of Tau nanoclusters, which subsequently evolve into cytotoxic assemblies in neuroblastoma cells. Using NMR spectroscopy and molecular dynamics (MD) simulations, we characterized the atomic-level impact of Aβ40 on Tau structure, dynamics, and interactions. We further demonstrate that Aβ40 enhances Tau phase separation, accelerates liquid-to-solid transitions, and promotes amyloid fibril formation. Together, these findings provide mechanistic insights into the Aβ-Tau interplay, offering a deeper understanding of Alzheimer's disease

pathogenesis and pointing toward potential therapeutic strategies targeting both LLPS and fibrillization processes.

## Results

### Aβ40 does not undergo LLPS

Aβ fibrillation has been widely studied, mostly with emphasis on aggregation kinetics and fibril structure[27–29]. However, it is not yet known whether Aβ undergoes LLPS in vitro. To evaluate the LLPS propensity of Aβ, we utilized the catGRANULE algorithm[30], which predicts phase separation potential of Aβ40 based on the primary sequence properties. The predicted LLPS propensity score of 0.248 indicates a low likelihood of phase separation. Aβ40 carries a net negative charge under physiological conditions, containing six negatively charged and three positively charged residues along its sequence (Supplementary Fig. 1A).

To experimentally assess Aβ40 phase separation, turbidity measurements were performed at varying protein concentrations in the presence of 10% and 15% PEG8000 as a crowding agent, with and without a 12-hour incubation period. As shown in Supplementary Fig. 1B–E, no significant increase in turbidity was observed under any tested condition, suggesting that Aβ40 does not undergo LLPS under the current experimental conditions. Further characterization using fluorescence microscopy with TAMRA-labeled Aβ40 (Supplementary Fig. 1F) revealed the formation of aggregates rather than liquid-like droplets, reinforcing this conclusion. To investigate the material properties of these aggregates, FRAP experiments were conducted (Supplementary Fig. 1G), demonstrating that Aβ40 formed solid-like aggregates rather than dynamic liquid-like condensates. Overall, our experimental observations show that, unlike Tau, Aβ40 does not form droplets under the tested conditions. This finding is consistent with catGRANULE predictions, which indicate a much lower phase separation propensity of Aβ40 compared to Tau, and is further supported by our coarse-grained simulations presented in the following sections.

### Aβ40 promotes Tau phase separation

The intrinsically disordered Tau exhibits a distinct multivalent charge distribution under physiological conditions (pH 7.4), as illustrated in Fig. 1A. Computational predictions using the catGRANULE algorithm yielded a droplet-promoting propensity score of 1.689 for Tau[30], consistent with previous studies confirming its ability to undergo phase separation[20,31].

To investigate the modulation of Tau LLPS by Aβ40, we systematically varied Aβ40 and Tau concentrations in the presence of 10% PEG8000 by using protein crystallization robotic dispenser and imager to generate a phase diagram (Fig. 1B & Supplementary Fig. 2). Compared to Tau alone, the presence of Aβ lowered the critical concentration required for Tau LLPS. Turbidity measurements at 350 nm further confirmed these findings, as Aβ40 alone did not increase turbidity, whereas its presence significantly enhanced Tau phase separation, resulting in a three-fold increase in absorbance compared to Tau droplets alone (Fig. 1C). These results were corroborated by brightfield microscopy images (Supplementary Fig. 2), which showed increased droplet formation in the presence of Aβ40.

To further investigate the co-condensation between Aβ40 and Tau, we fluorescently labeled both proteins and monitored their incorporation using fluorescence microscopy. Alexa-647-labeled Tau and TAMRA-labeled Aβ40 exhibited co-localization within the Tau droplets, indicating the recruitment of Aβ40 into Tau condensates (Fig. 1D). The fusion dynamics of these droplets confirmed their liquid-like properties, as evidenced by characteristic fusion events (Fig. 1E). To investigate the effect of Aβ40 on Tau droplet growth, we monitored the size of Tau droplets over time. Representative time-lapse microscopy images (Supplementary Fig. 3A) show that when Tau was co-incubated with Aβ40, the formation of droplets showed a more rapid increase, with an increased number of larger droplets forming an

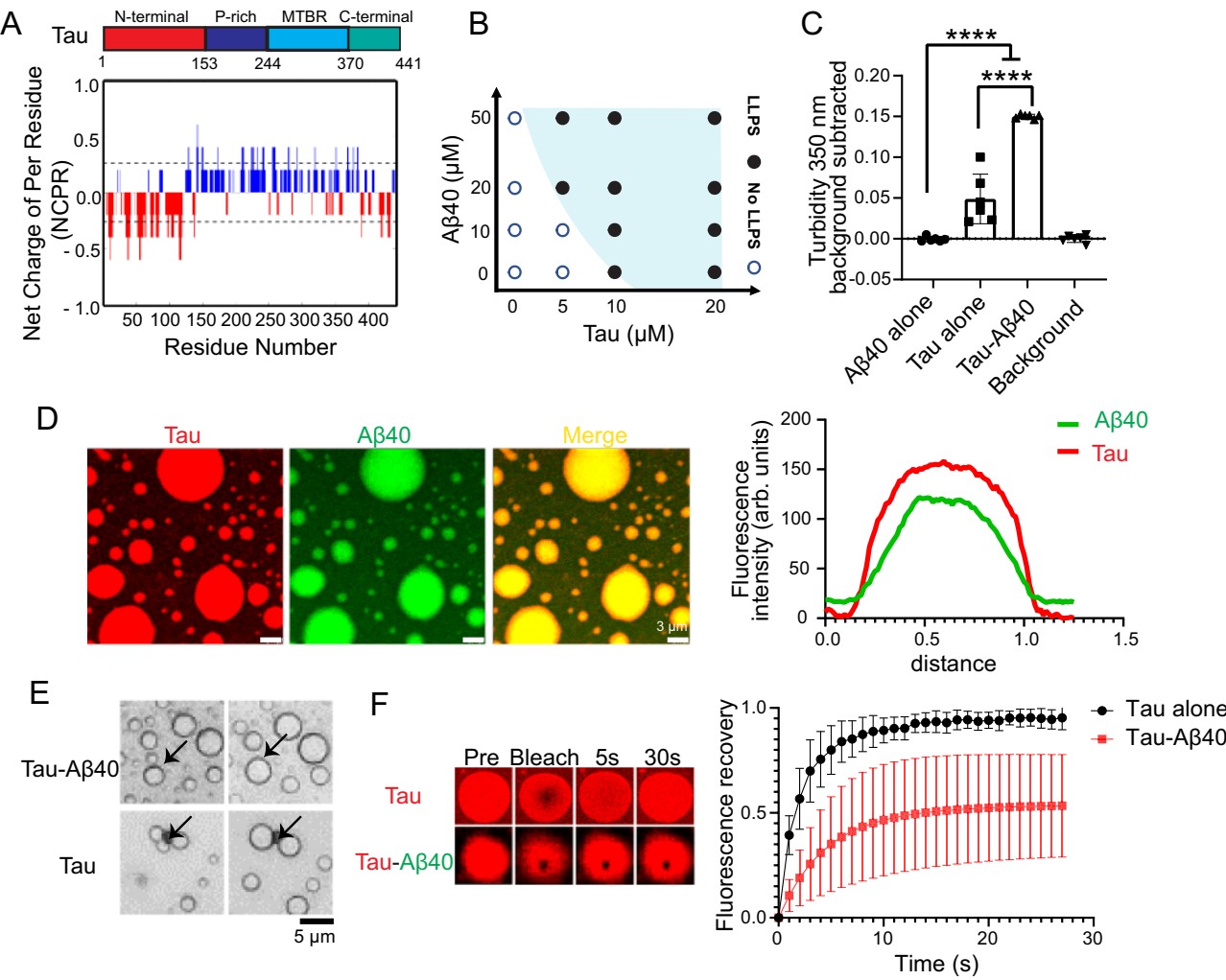

**Fig. 1 | Aβ40 promotes Tau LLPS in vitro. A** Up panel: Schematic representation of Tau (2N4R, Uniport P10636-8) protein sequence. The highly disordered N-terminal domain (red), proline-rich regions (dark blue), MTBR (microtubule binding region, light blue) and C-terminal domain (green). Below panel: Net charge per residue (NCPR) distribution of Tau sequence revealed the negatively charged (red) N-terminal domain, the positively charged (blue) middle domain and the negatively charged (red) C-terminal domain. NCPR was calculated at the CIDER server[30] (http://pappulab.wustl.edu/CIDER/). **B** Phase diagram corresponding to images (Supplementary Fig. 2) of Tau LLPS versus Aβ40 concentration in the presence of 10 % PEG. **C** Turbidity measurements (350 nm) of freshly prepared 10 μM Tau in the presence of Aβ40 (molecular ratio 1:1). The data represent the mean ± SEM. (*n* = 6 independent biological samples, A paired t-test was used, **** *P* < 0.0001). **D** Left: Representative fluorescence images of 10 μM Tau (Alexa-647 labelled) phase-separated droplets formation in the presence of 10 μM Aβ40 (TAMRA labelled) and 10% of PEG. Right: Fluorescence intensities of Aβ40-TAMRA and Tau-Alex647 indicates the co-localization of Aβ40 and Tau. **E** Liquid-like droplet fusion event of 10 μM Tau with or without 10 μM Aβ40. **F** Representative FRAP images (left) and analysis (right) of 10 μM Tau with or without 10 μM Aβ40. The data represent the mean ± SEM. (*n* = 3 independent biological samples). Buffer condition: 50 mM PB, 150 mM NaCl (pH 7.4), 10% PEG8000.

earlier time point compared to Tau alone. The observed crossover in droplet growth kinetics (Supplementary Fig. 3B) where Tau+Aβ40 droplets initially form and expand faster than Tau-only droplets but later exhibit slower growth. This reflects an early Aβ40-driven enhancement of condensate nucleation and maturation followed by reduced droplet mobility and Tau monomer depletion due to Aβ-induced liquid–solid transition and fibril formation, whereas Tau-only samples continue to enlarge through prolonged coalescence and Ostwald ripening. At late stage, the droplet diameters with or without Aβ40 reached comparable values, indicating that Aβ40 primarily influences the kinetics of phase separation rather than the final equilibrium state. Overall, Aβ40 incorporation promoted droplet maturation, leading to the formation of larger condensates.

To determine whether Aβ40 alters the material properties of Tau droplets, we conducted FRAP experiments (Fig. 1F & Supplementary Fig. 4). Freshly formed Tau droplets exhibited nearly full fluorescence recovery, consistent with a dynamic liquid-like state. However, in the

presence of Aβ40, Tau fluorescence recovery was significantly reduced to approximately 50% of the initial intensity, indicating a slower molecular diffusion within the droplets and a transition toward a more gel-like or solid-like state. These findings demonstrate that Aβ40 is recruited into Tau droplets, enhances Tau phase separation and dynamical arrest.

### Aβ40 accelerates Tau phase transition
Intrinsically disordered proteins (IDPs), such as TDP-43 and FUS, form liquid droplets that gradually evolve into more viscous, gel-like states and eventually transition into amyloid-like or amorphous aggregates[25,32–34]. To determine whether Tau undergoes a comparable maturation process, progressing from a liquid phase to a gel-like and ultimately an aggregated state, we did long-term time lapse imaging and analyzed the dynamical arrest of Tau droplets over time.

To evaluate the impact of Aβ40 on Tau phase transition over time, fluorescence microscopy was used to visualize condensates formed by

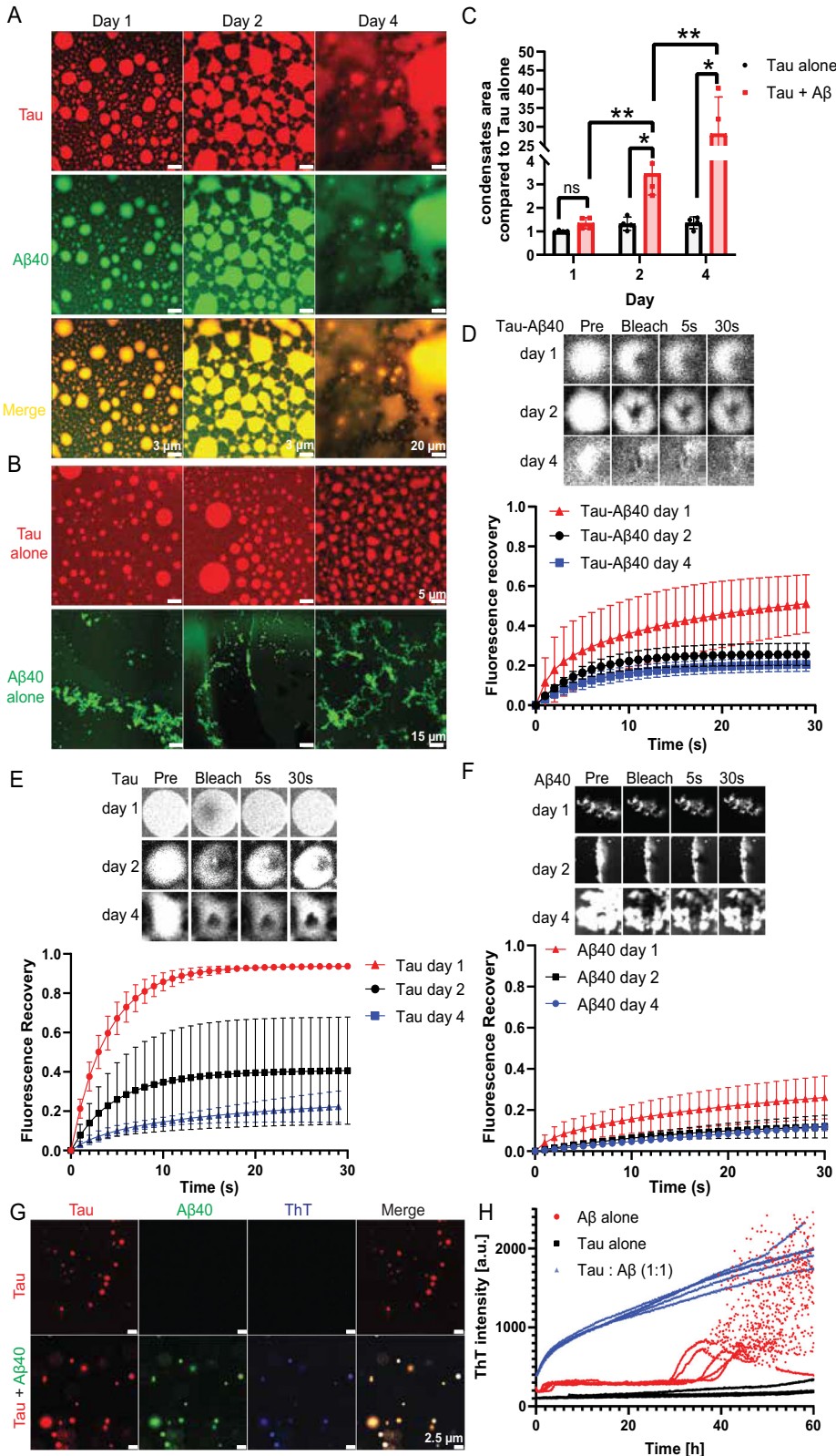

10 µM Tau in the presence and absence of 10 µM Aβ40 (Fig. 2A-B). In the presence of Aβ40, Tau initially formed spherical droplets, which gradually transitioned into star-like droplet network by day 2, finally formed irregular and less dynamic structures by day 4. In contrast, Tau alone maintained stable phase-separated spherical droplets until day 2 then transitioned to the droplet network by day 4, while Aβ40 alone exhibited an aggregated, solid-like morphology over time.

Quantitative analysis of the total condensate area revealed a significant increase in Tau-Aβ40 structures compared to Tau alone (Fig. 2C). By day 2 and day 4, the total condensate area of Tau-Aβ40 mixtures was significantly larger than Tau alone, indicating an accelerated phase transition in the presence of Aβ40. To assess the material properties of these condensates, FRAP experiments were performed (Fig. 2D–F). Tau alone exhibited rapid fluorescence recovery on day 1, suggesting a

**Fig. 2 | Aβ40 accelerates Tau phase transition and fibrillation in vitro.**
**A** Representative images of phase separated structures formed by 10 μM Tau with
Aβ40 after 1, 2 and 4 days incubation at room temperature. Scale bar is presented.
**B** Representative images of phase separated structures formed by 10 μM Tau alone
(top) and aggregated structures formed by Aβ40 alone (bottom) after 1, 2 and
4 days incubation at room temperature. Scale bar is presented. **C** Quantification of
phase-separated and aggregated fluorescence structures formed by Tau with or
without Aβ40 (total area). The data represent the mean ± SEM. ($n$ = 4 independent
biological samples, a two-sided unpaired Student's t-test was used, day 2:
$p$ = 0.0153; day 4: $p$ = 0.0127; Tau + Aβ day 1 vs Tau + Aβ day 2: $p$ = 0.0042; Tau + Aβ

day 2 vs Tau + Aβ day 4: $p$ = 0.0023; ns: not significant, *$p$ < 0.05, **$p$ < 0.01).
Representative FRAP images (top) and analysis (bottom) of Tau with Aβ40 (**D**), Tau
alone (**E**) and Aβ40 alone (**F**). The data represent the mean ± SEM. ($n$ = 3 independent
biological samples). **G** Representative ThT staining images of Tau alone and
Tau with Aβ40 after 1 h room temperature incubation. Buffer condition: 20 μM ThT,
50 mM PB, 150 mM NaCl (pH 7.4). Scale bar is presented. **H** Fibrillation assay of Tau
with and without Aβ40. 15 μM Tau was incubated in the presence of 10% PEG8000
at 37 °C under shaking conditions with or without 15 μM Aβ40. ($n$ = 5 independent
biological samples).

dynamic, liquid-like state (Fig. 2E). However, by day 4, the fluorescence
recovery was significantly reduced, indicating progressive maturation
into a gel-like or solid-like state. In contrast, Tau-Aβ40 condensates
displayed markedly reduced fluorescence recovery at all time points,
with an even lower recovery rate observed by day 4 (Fig. 2D). This
suggests that Aβ40 accelerates the transition of Tau condensates from
a liquid-like to a more rigid, less dynamic state and inhibits the mole-
cule exchange of droplets. This maturation process closely parallels
the phase transition observed in phosphorylated Tau, which exhibits a
gradual reduction in molecular exchange over time[20]. Aβ40 alone
showed minimal fluorescence recovery throughout the incubation
period, consistent with the formation of solid-like aggregates rather
than phase-separated droplets (Fig. 2F). Taken together, these results
demonstrate that Aβ40 promotes and accelerates the phase transition
of Tau condensates, leading to a loss of dynamic properties and the
formation of more rigid, aggregated structures over time.

### Aβ40 potentiates Tau fibrillation in vitro

The process of Tau aggregation into fibrils has been extensively
characterized, with numerous studies focusing on the kinetics of
aggregation and the structural properties of the resulting fibrillar
assemblies[35,36]. A predominant approach in these investigations has
involved the use of polyanionic co-factors, like heparin, to induce fibril
formation[37,38]. To eliminate the potential confounding effect of co-
factors competing with Aβ40 in modulating Tau fibrillation, we
employed confocal imaging and aggregation kinetics assays con-
ducted in the absence of co-factors and in the presence and absence of
Aβ40 (Fig. 2G-H). We utilized Thioflavin T (ThT) fluorescence to
monitor Tau aggregation, which is widely used to detect amyloid-like
protein aggregates[39].

To assess the influence of Aβ40 on Tau fibrillation within a phase-
separated droplet state, we conducted ThT staining (Fig. 2G) and
fibrillation kinetics (Fig. 2H) in the presence of 10% PEG8000. There is
no ThT positive signal observed in Tau alone sample. However, a
strong ThT signal was detected only in the co-condensate of Tau with
Aβ40, indicating the presence of β-sheet-rich amyloid aggregates
(Fig. 2G). Similarly, in the fibrillation kinetics, the co-incubation of Tau
and Aβ40 (blue) resulted in a significant increase in ThT fluorescence
and no lag phase, suggesting that Aβ40 facilitates Tau fibrillation in the
droplet state (Fig. 2H). Together, these results demonstrate that Aβ40
accelerated Tau fibrillation within droplets which serve as a precursor
for Tau aggregation.

### Molecular insights into the interaction between Tau and Aβ40

Molecular interactions play a crucial role in driving the phase separa-
tion and aggregation of Tau and other IDPs[20,24,25]. To gain high-
resolution insights into the interaction between Tau and Aβ40 at
monomeric or condensate states, we performed 2D $^1$H-$^{15}$N hetero-
nuclear single quantum coherence (HSQC) NMR spectroscopy using
100 μM isotopically labeled $^{15}$N-Tau in the presence and absence of
Aβ40 and compared the cross-peak intensities and chemical shift
perturbation (CSP) (Fig. 3). The experiments were conducted at 283 K
to minimize proton signal loss due to solvent exchange[40]. In the
absence of PEG, equimolar addition of Aβ40, which did not form

droplets confirmed via microscopy, resulted in a global increase in
cross-peak intensity by approximately 30%-40% (Fig. 3A-C). This
observation suggests that Aβ40 shifts the equilibrium of Tau by dis-
solving NMR-invisible Tau nanoclusters. This observation is further
supported by mass photometry analysis (Supplementary Fig. 6). Aβ40
at both 5 μM and 25 μM formed small nanoclusters with predominant
mass distributions at ~90 kDa and ~84 kDa, respectively. Minor higher-
order species were observed around 149–150 kDa, but overall Aβ40
exhibited limited multimerization for nanocluster formation under
these conditions. In contrast, Tau showed a broader distribution of
higher molecular weight nanocluster species. Tau formed nanoclus-
ters with peaks at 123 kDa, and 182 kDa, indicating enhanced multi-
merization and nanocluster formation. Upon co-incubation of 5 μM
Tau with increasing concentrations of Aβ40 (2.5–25 μM), a notable
reduction in Tau nanocluster size and abundance was observed. The
resulting mass distributions showed significantly diminished higher-
order assemblies compared to Tau alone. These data suggest that
Aβ40 interferes with or dissolves large Tau nanoclusters, likely
through competitive interactions or hetero-oligomer formation that
impedes Tau self-association. These findings are consistent with our
previous studies using $^{15}$N-Aβ NMR measurements, which demon-
strated that Tau solubilizes Aβ oligomers[12]. Notably, only minimal
CSP were observed, with slight perturbations detected in the PHF6*
(residues 275–280) region. However, these structural and global con-
formational changes were not detectable by CD spectroscopy (Sup-
plementary Fig. 7). To further obtain insights into the interaction
between Tau and Aβ40 at condensate state, 2D $^1$H-$^{15}$N HSQC NMR
spectroscopy was performed. The addition of 10% PEG resulted in the
droplet formation, as confirmed by microscopy. Residue-specific
analysis showed minor CSP (Supplementary Fig. 5). The minimal
CSP likely result from conformational compaction within the droplet
state induced by excluded volume effects from PEG, rather than direct
interaction[41]. Upon the equimolar addition of Aβ40 in the presence of
10% PEG, cross-peak intensities exhibited a global increase of
approximately 50% (Fig. 3D-F). This suggests that PEG partially con-
tributes Aβ40-induced dissolution of Tau nanoclusters. Moreover,
Aβ40 induced enhanced CSP in multiple regions, particularly PHF6*,
the R4 domain (residues 340–370), and the broader N-terminal region
(Fig. 3E). These observations imply enhancement of Aβ40–Tau inter-
actions by increasing the effective local concentration and facilitating
transient contacts, thereby promoting condensate formation. Fur-
thermore, crowding effects likely contribute to reduced NMR inten-
sities by stabilizing certain bound states or slowing conformational
exchange dynamics. Collectively, our findings suggest that while Aβ40
and Tau interact with similar binding patterns in both aqueous and
condensate phases, their interactions are stronger within the con-
densate environment.

### Molecular mechanisms underlying the enhancement of Tau
### phase separation by Aβ40

Molecular dynamics (MD) simulations have been widely used to elu-
cidate the mechanisms underlying protein phase separation and the
phase behaviors[42–47]. To investigate how Aβ40 solubilizes Tau and then
enhances Tau phase separation at the molecular level, we conducted

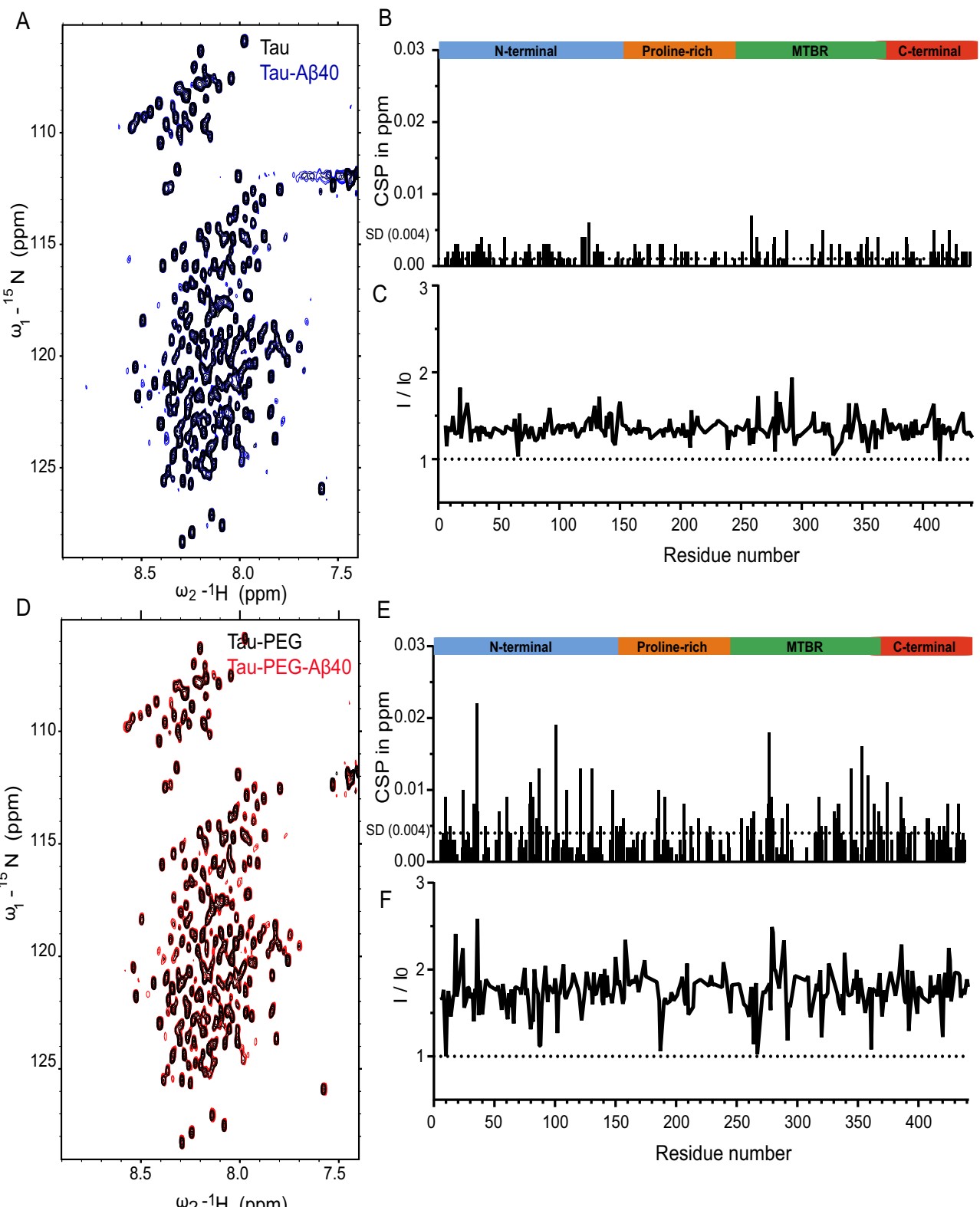

**Fig. 3 | Aβ40 interacts with Tau monomers. A**–**C** 2D NMR $^1$H-$^{15}$N HSQC experiments with 100 μM $^{15}$N-labeled monomeric Tau in the absence (black) and presence of Aβ40 (molecular ratio 1:1, green). The experiments were performed in 20 mM sodium phosphate (pH 6.3). The chemical shift perturbations (**B**) and the signal intensity changes (**C**). **D**–**F** 2D NMR $^1$H-$^{15}$N HSQC experiments with 100 μM $^{15}$N-labeled monomeric Tau in the absence (black) and presence of Aβ40 (molecular ratio 1:1, red) with 10% PEG 8000 at the droplet state. The chemical shift perturbations (**E**) and the signal intensity changes (**F**).

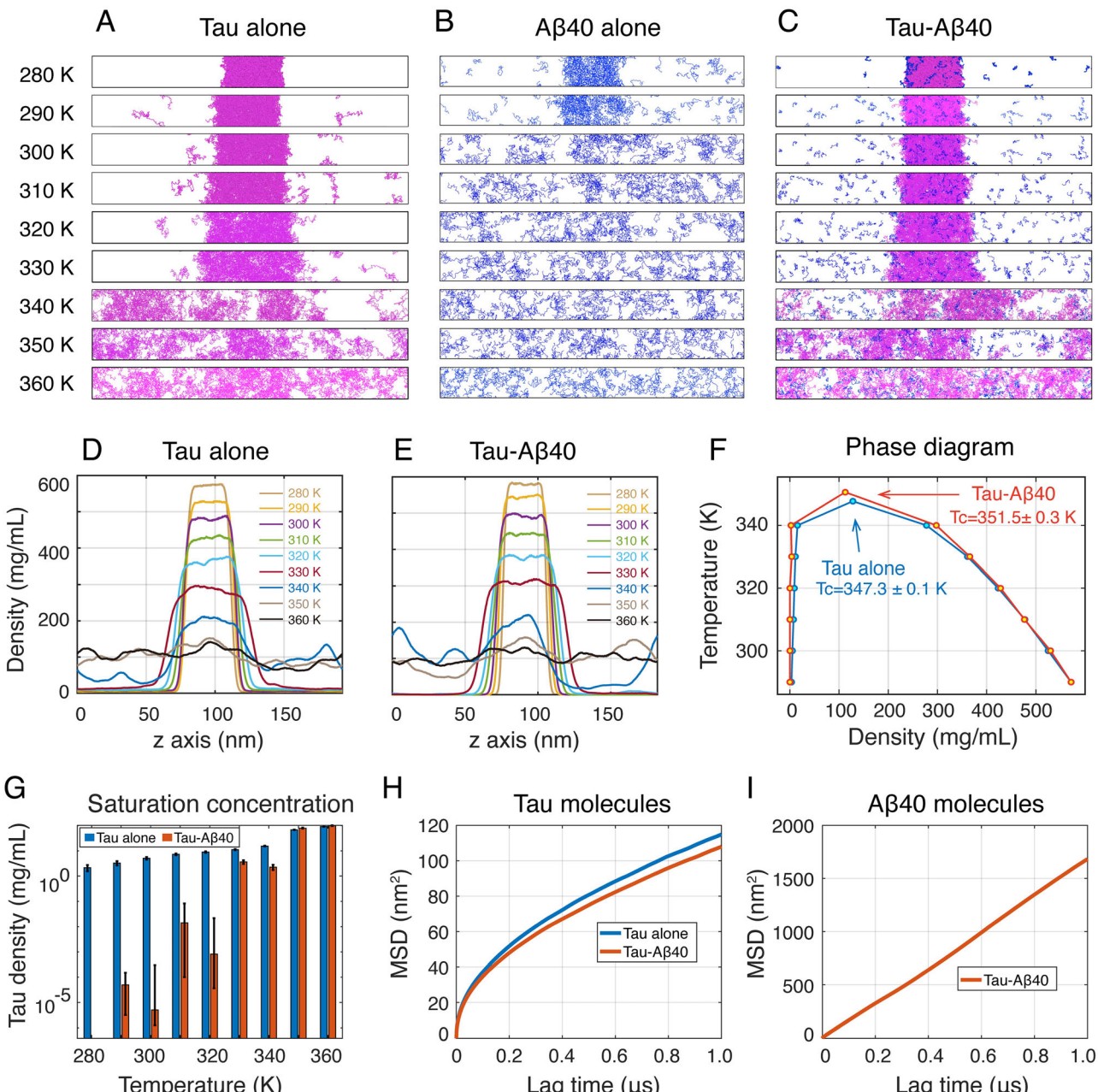

**Fig. 4 | Phase separation propensity and phase behavior of Tau in the absence and presence of Aβ40.** A–C Representative simulation snapshots of Tau alone (**A**), Aβ40 alone (**B**) and Tau-Aβ40 (**C**) systems at nine different temperatures. **D**, **E** Density profiles along the z-axis at nine temperature points for the Tau alone (**D**) and Tau-Aβ40 (**E**) systems. **F** Phase diagrams of the Tau and Tau-Aβ40 systems with critical temperatures obtained by fitting to critical equation. Error bars for the dense and dilute phase densities represent the 95% confidence intervals of the fitted density values for the two phases. Error bars for the critical temperature correspond to the 95% confidence intervals of the fitted critical temperature values. **G** The saturation concentration of Tau molecules of the Tau alone and Tau-Aβ40 systems. Mean values are obtained by averaging the density profiles for each system and the error bars indicate the standard deviation across density profiles. **H, I** MSD of Tau molecules within Tau alone and Tau-Aβ40 condensates (**H**) and that of Aβ40 molecules within Tau-Aβ40 condensate.

MD simulations on three systems: Tau alone, Aβ40 alone, and a mixture of Tau and Aβ40 (Tau-Aβ40). Each simulation began from a phase coexistence state, featuring a preformed dense phase and an initially empty dilute phase, as done in previous studies[48–50].

Tau alone demonstrated a strong propensity for phase separation across a broad temperature range (280-330 K), with a gradual decline as temperature increased (Fig. 4A). At 340 K, phase separation dropped sharply compared to at 330 K and was completely lost above this temperature. In contrast, Aβ40 only phase separate at very low (<300 K) temperature and did not exhibit phase separation at 300 K or higher temperatures (Fig. 4B). When both proteins were present, they

co-phase separated into a dense phase containing both Tau and Aβ40, while a significant portion of Aβ40 were distributed in the dilute phase (Fig. 4C). Increasing temperature weakened the phase separation capability of the Tau-Aβ40 system, with complete mixing occurring above 340 K (Fig. 4C). We observed that, below 340 K, the dense phase formed by the Tau-Aβ40 mixture was more compact than that formed by Tau alone at each temperature, indicating a higher dense-phase density. This observation is further supported by density profile analyses as described below (Fig. 4D-E). In the Tau alone system, the density plateau progressively declined with temperature increasing, alongside an increase in dilute-phase density. At temperatures above

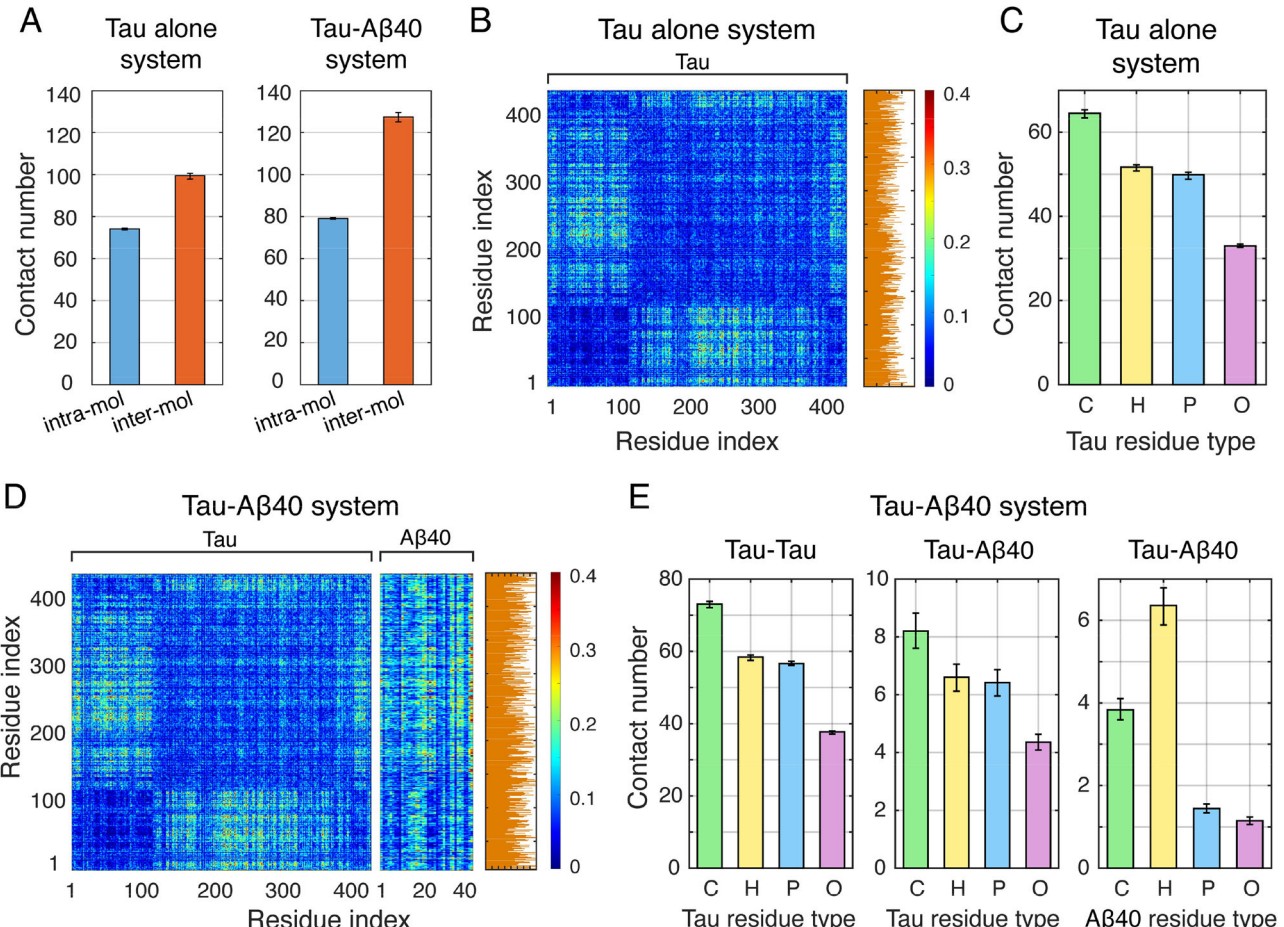

**Fig. 5 | Molecular interactions driving the phase separation of Tau in the absence and presence of Aβ40. A** Intra-molecular and inter-molecular contact numbers in the Tau system and Tau-Aβ40 systems. The error bars represent the minimum and maximum contact numbers obtained from four equal segments of the final 1 μs of each trajectory (*n* = 4). The raw data from these four segments are displayed as dot plots adjacent to the bars. The same representation is used for all subsequent bar plots in this figure. **B** Inter-molecular contact numbers between each pair of Tau residues in the Tau system. The accumulated contribution of each Tau residue to inter-molecular contacts is shown on the right. **C** Accumulated

contact numbers of each Tau residue type with all residues, i.e., charged (C), hydrophobic (H), polar (P), and other (O) residues, in the Tau alone system. **D** Inter-molecular contact numbers between each Tau residue and each Tau/Aβ40 residue in the Tau-Aβ40 system. The accumulated contribution of each Tau residue to inter-molecular contacts is shown on the right. **E** Accumulated contact numbers of each pair of residue types in the Tau-Aβ40 system. Contributions of each Tau residue type in Tau-Tau interaction, each Tau and Aβ40 residue type in Tau-Aβ40 interaction.

340 K, this plateau disappeared, resulting in a uniform density profile consistent with a complete loss of phase separation (Fig. 4D). A similar trend was observed for the Tau-Aβ40 system, with consistently, albeit slightly, higher dense-phase densities than Tau alone system at all temperatures, suggesting an enhancement in phase separation propensity (Fig. 4F). Fitting the density profiles to the critical equation yielded upper critical solution temperature (UCST) of 347.3 K for the Tau system and 351.5 K for the Tau-Aβ40 system (Supplementary Table 1). This difference in critical temperature (4.2 K) is comparable to those reported in previous coarse-grained simulations assessing differences in phase separation capability between protein systems[51,52]. For example, the D290V mutation, which experimentally reduces the phase separation capability of hnRNPA2, leads to a ~1% (~3 K) decrease in critical temperature in HPS simulations[5]. Removal of a C-terminal β-sheet-rich motif of α-synuclein results in a ~5 K decrease in critical temperature of phase separation in HPS simulations[52]. We then calculated the saturation concentration of Tau to further quantify the impact of Aβ40 on the phase separation capability of Tau. This metric has been widely used to evaluate the phase separation capabilities of protein systems[53–56]. Our calculations show that the saturation concentration of Tau molecules in the presence of Aβ40 is markedly lower than in the absence of Aβ40 near physiological

temperature (290-320 K), indicating that Aβ40 enhances the phase separation propensity of Tau. To assess the dynamics of the condensates, we followed previous studies[48,57] and extracted the dense phases from the final state of the phase-coexistence simulations and placed them into cubic simulation boxes (Supplementary Fig. 8). The dense phases were then simulated for 5 μs at 310 K in the NVT ensemble. Using the last 1 μs of each trajectory, we calculated the mean square displacement (MSD) for Tau molecules in the Tau-alone and Tau-Aβ40 systems, as well as the MSD for Aβ40 molecules in the Tau-Aβ40 system. The MSD values for Tau molecules display a non-linear behavior with lag time, suggesting sub-diffusion behaviors (Fig. 4H). Fitting MSD curves to power-law diffusion equation $MSD = K_\alpha \tau^\alpha$ yields a smaller diffusion exponent for Tau molecules within the Tau-Aβ40 condensate (0.494 ± 0.001) compared to those within Tau-alone condensate (0.500 ± 0.001), indicating that the presence of Aβ40 reduces the fluidity of Tau molecules within the condensate. Furthermore, the MSD of Aβ40 in the Tau-Aβ40 system is significantly larger than that of Tau and follows a diffusive behavior characterized by the Stokes-Einstein relation ($MSD = 6D\tau$) with a diffusive coefficient (D) of 1.696 ± 0.004 nm²/ns (Fig. 4I). These observations suggest that Aβ40 molecules remain highly dynamic within the condensate while reducing the dynamics of Tau molecules,

thereby providing further support for the Aβ40-induced acceleration of Tau phase transition observed in our experiments.

To elucidate the molecular forces governing the Aβ40-enhanced Tau phase separation, we performed interaction analyses on Tau alone and Tau-Aβ40 systems. In both systems, the number of intermolecular contacts was much larger than that of intramolecular contacts (Fig. 5A), and intramolecular contacts predominantly occurred between residues in close sequence proximity (Supplementary Fig. 9). The presence of Aβ40 led to a marked enhancement of intermolecular contacts but a slight increase of intramolecular contacts (Fig. 5A), suggesting that Tau promotes intermolecular interactions within the Tau-Aβ40 system.

In the Tau alone system, strong intermolecular interactions were observed between residues 40–120 in the N-terminal region and residues 120–400 containing the 4 R repeat domain, residues 244–369 (Fig. 5B) which are respectively enriched in negatively and positively charged residues (Supplementary Fig. 10A). This suggests a key role of electrostatic attraction in Tau LLPS. Residue-accumulated contact numbers revealed broad contributions from many residues, including both charged residues and those with high hydrophobicity scores (Fig. 5B, Supplementary Figs. 11, 10B), indicating the involvement of multiple interaction types. To further dissect these contributions, we calculated the accumulated contact numbers of different residue types. Our calculations indicated that charged residues were the primary drivers of Tau phase separation, with hydrophobic and polar residues also making non-negligible contributions (Fig. 5C). The significance of polar residues in promoting phase separation has also been reported in a recent study on FUS, another intrinsically disordered protein[58].

In the Tau-Aβ40 system, the inter-Tau contact pattern closely resembled that observed in the Tau alone system but showed increased contact numbers across most residue pairs (Fig. 5D & Supplementary Fig. 11), suggesting that the presence of Aβ40 globally enhanced intermolecular interactions between Tau molecules, in particular the interactions between the 4 R repeat domain and the N-terminal region. We further analyzed the interactions between Tau and Aβ40. High contact numbers were observed for hydrophobic-residue-rich regions of Aβ40, i.e., residues 15-22 ($_{15}$QKLVFFAE$_{22}$) and residues 30-40 ($_{30}$AIIGLMVGGVV$_{40}$) (Figs. 5D, Supplementary Fig. 12). Analysis of accumulated contact numbers for each residue type showed that charged residues of Tau protein dominate Tau-Aβ40 interactions followed by polar and hydrophobic residues, while hydrophobic residues of Aβ40 dominate the Tau-Aβ40 interaction followed by charged residues (Fig. 5E). In addition, intermolecular interactions between Aβ40 molecules were relatively weak (Supplementary Fig. 13), suggesting a minimal contribution to phase separation. Together, these results demonstrate that Aβ40 interacts with Tau mainly through an interplay of hydrophobic and electrostatic interactions.

Collectively, our simulations revealed that, Aβ40 engages with Tau through its charged and hydrophobic residues, thereby promoting Tau-Tau intermolecular electrostatic interactions, and, to a lesser extent, hydrophobic and polar interactions. These combined effects ultimately enhance the phase separation of Tau.

## Aβ40 promote Tau induced toxicity

To further explore how Aβ40 affects the toxicity of Tau aggregates in these environments, we used the A11 antibody, which specifically detects toxic amyloid aggregates with β-sheet structures[59]. For Aβ40 alone, increasing its concentration resulted in more intense sample dots (Fig. 6A), which indicates higher toxicity. A similar trend was observed in SH-SY5Y cell viability assays, where higher Aβ40 concentrations were associated with increased toxicity (Fig. 6C). When Tau was co-incubated with Aβ40, the resulting aggregates showed a slight increase in toxicity compared to Tau alone, although the

difference was not statistically significant (Fig. 6B). Interestingly, when the Tau: Aβ40 ratio increased, the overall toxicity remained relatively unchanged (Fig. 6B). At the cellular level, Aβ40-Tau co-aggregates at a 1:1 molar ratio displayed similar toxicity to Tau alone. Surprisingly, as the Aβ40 proportion increased, a slight reduction in toxicity was observed (Fig. 6C). This suggests that Aβ40 may partially disassemble Tau aggregates into monomers, thereby reducing their cytotoxic potential. Then we used fluorophore-labelled proteins to check if these proteins enter in cells. We found that co-incubation with Tau increased intracellular Aβ40 signal (Supplementary Fig. 14A-B), which parallels the higher cytotoxicity observed for the mixture compared with Aβ40 alone. Both Aβ and Tau have been reported damage cellular membranes and promote internalization[60–62]. One possible explanation is that Aβ40 renders Tau more soluble, promoting Tau to associate with the cell membrane to form oligomers. These oligomers then insert into the lipid bilayer, increase membrane permeability, and in turn facilitate Aβ40 uptake. Further studies are required to validate this proposed mechanism.

Given that the cellular environment is inherently crowded, 10% PEG 8000 was used to mimic artificial cellular environment in vitro, as described in previous studies[63]. Under this crowding condition, Aβ40 aggregates became more toxic in a concentration-dependent manner, as shown by both dot blot analysis and cell viability assays (Fig. 6E & F). More importantly, when Tau was co-incubated with Aβ40 under crowding condition, the resulting aggregates exhibited threefold higher toxicity than Tau alone (Fig. 6E). Additionally, as the Tau-Aβ40 ratio increased, the toxicity increased further. Consistently, in SH-SY5Y cells, Aβ40-Tau condensates exhibited higher toxicity compared to Tau alone (Fig. 6F). Due to the high toxicity of the co-condensates, cell toxicity remains comparable as the Tau-Aβ40 ratio increased. These findings suggest that Aβ40 enhances Tau toxicity by promoting its phase separation, leading to the formation of more toxic aggregates with more rigid structures, consistent with a previous study[64].

In summary, LLPS plays a critical role in modulating Aβ40 and Tau co-aggregation toxicity. In the absence of PEG, Aβ40 appears to partially dissolve Tau aggregates into monomers, thereby reducing co-aggregate toxicity. However, in the presence of PEG, Aβ40 accelerates Tau LLPS, resulting in the formation of highly toxic aggregates. These results indicate that Aβ40 and Tau follow distinct co-aggregation pathways under crowding versus uncrowding conditions. This aligns with our previous observations, which further support the role of Aβ40 in modulating Tau aggregation and toxicity under different conditions.

## Discussion

In the brains of AD patients, the colocalization of Aβ plaques and Tau tangles is commonly observed[65,66]. While these lesions represent the end stage of protein aggregation, growing evidence suggests that the interplay between Tau and Aβ plays a more central role in disease progression[67]. Aβ and Tau synergistically promote each other's pathology both in vitro and in vivo, with Aβ suggested to act upstream, triggering a cascade that ultimately leads to Tau pathology[12,68,69]. To better understand the disease-related consequences of this cross-interaction, it is crucial to elucidate the interactions between Aβ and Tau.

Our results indicate that monomeric Aβ40 does not undergo phase separation under the experimental conditions (Supplementary Fig. 1). However, several studies have reported that Aβ can exhibit LLPS under certain conditions[17,18,70]. Consistently, our coarse-grained simulations show that Aβ40 is capable of phase separation only at low temperatures (≤290 K) but loses this ability at higher temperatures. These findings suggest that Aβ40 possesses an intrinsically weak phase separation propensity, which is highly sensitive to environmental conditions. This underscores the crucial role of experimental conditions, protein variants, and external modulators in influencing protein behavior. Previous studies have reported that Aβ42 oligomers can

 

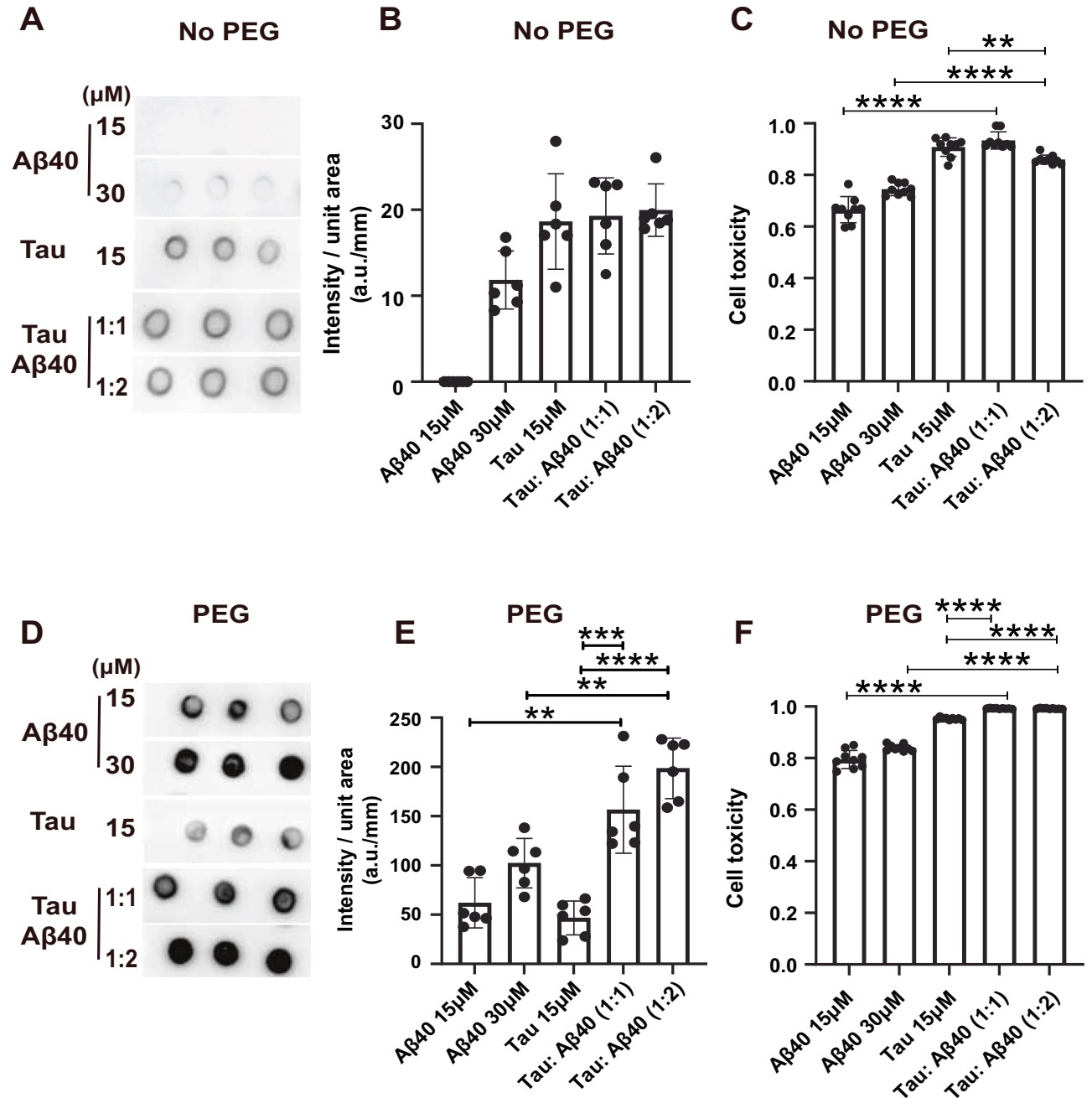

**Fig. 6 | Aβ40 differentially regulates Tau induced toxicity at aggregates and condensates stages.** Dot blot assay (**A**) and quantification (**B**) of varied concentration of Aβ40, Tau, Tau: Aβ40 at 1:1 (15 μM:15 μM) and 1:2 (15 μM:30 μM) molar ratios. All samples were incubated at 37 °C for 2 h in the absence of 10% PEG8000. The data represent the mean ± SEM. *n* = 6 independent biological samples. **C** SH-SY5Y cell viability under various conditions, including different concentrations of Aβ40, Tau, Tau: Aβ40 at 1:1 and 1:2 molar ratios in the absence of 10% PEG8000. The data represent the mean ± SEM. Statistical significance was determined by a two-sided unpaired Student's t-test. Tau 15 μM vs Tau:Aβ40 (1:2), *p* = 0.0023, ****p < 0.0001. *n* = 9 independent biological samples. Dot blot assay (**D**) and quantification (**E**) of varied concentration of Aβ40, Tau, Tau: Aβ40 at 1:1

(15 μM:15 μM) and 1:2 (15 μM:30 μM) molar ratios. All samples were incubated at 37 °C for 2 h in the presence of 10% PEG8000. The data represents the mean ± SEM. Statistical significance was determined by a two-sided unpaired Student's t-test, Aβ40 15 μM vs Tau:Aβ40 (1:1), *p* = 0.0011; Aβ40 30 μM vs Tau:Aβ40 (1:2), *p* = 0.0001; Tau 15 μM vs Tau:Aβ40 (1:1), *p* = 0.0002, ****p < 0.0001. *n* = 6 independent biological samples. **F** SH-SY5Y cell toxicity under various conditions, including different concentrations of Aβ40, Tau, Tau: Aβ40 at 1:1 and 1:2 molar ratios in the presence of 10% PEG8000. All samples were pre-incubated at 37 °C for 2 h, prior to a 48-hour incubation with cells. Buffer: 50 mM PB, 150 mM NaCl, pH 7.4. The data represent the mean ± SEM. Statistical significance was determined by a two-sided unpaired Student's t-test, ****p < 0.0001. *n* = 9 independent biological samples.

phase separate under in the presence of SDS or lipid environment, forming liquid-like droplets that facilitate fibril formation[17,70]. The key differences between our study and these reports likely arise from variations in Aβ isoforms (Aβ40 vs. Aβ42), oligomeric states, and the presence of cofactors. Notably, Aβ42 has a higher aggregation propensity than Aβ40 due to its additional two C-terminal residues, which enhance hydrophobic and π-stacking interactions, potentially increasing its likelihood of undergoing LLPS[71].

Tau is an IDP that undergoes LLPS, forming dynamic, liquid-like condensates that may serve as precursors to aggregation[20]. Our

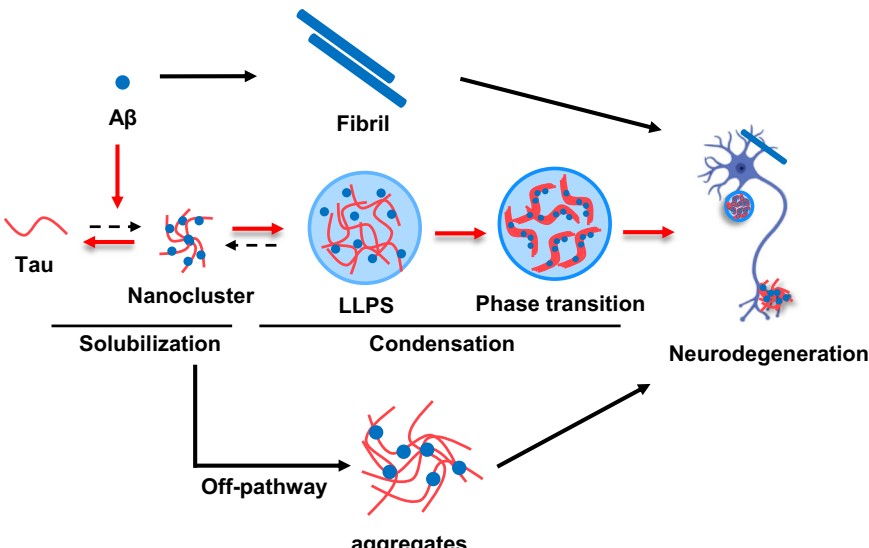

**Fig. 7 | Schematic illustration of the proposed mechanism of Aβ40 regulating Tau phase separation/aggregation behavior.** Based on our observations, we propose a model in which Aβ40 modulates the phase behavior of Tau. Initially, Aβ40 interacts with Tau to solubilize Tau nanocluster species, thereby shifting the equilibrium toward a monomeric state both in the presence and absence of macromolecular crowding and later forming the amorphous aggregates by off-pathway and damaging the neuron. However, upon incubation under crowded conditions, the nature of the interaction shifts. Tau and Aβ co-assemble into condensates by intermolecular electrostatic and hydrophobic interaction, ultimately promoting a phase transition toward solid-like aggregates further contribute to the neurodegeneration progression. The neuron model was created in BioRender. Luo, J. (2026) https://BioRender.com/4a5kqt2.

findings reveal that although Aβ40 does not undergo LLPS independently in solution, it efficiently partitions into Tau condensates, enriching within these droplets through interactions with multiple Tau regions, including PHF6, PHF6*, and the microtubule-binding domain. These interactions are mediated by a combination of electrostatic and hydrophobic forces, suggesting a client–scaffold relationship in which Aβ40 behaves as a recruited client and Tau functions as the scaffold[72]. This enrichment positions Aβ40 as a key modulator of Tau's phase behavior. Consistent with prior reports that other amyloidogenic proteins, such as α-synuclein, are recruited into Tau condensates via domain-specific interactions[73], our results support a broader model of amyloid cross-interactions contributing to the overlapping pathology of neurodegenerative disorders. The ability of Tau droplets to concentrate Aβ40 highlights their potential role as reservoirs that enhance local Aβ40 concentration, thereby facilitating intermolecular encounters that may initiate or accelerate aggregation pathways. These observations underscore the relevance of LLPS as a spatially and temporally organized platform for pathological protein interactions in AD.

The phase transition of condensates into aggregates is a defining feature of AD pathology[19]. Our findings that Aβ40 modulates the biophysical properties of Tau condensates by accelerating their phase transition from a dynamic, liquid-like state to a rigid, gel-like phase. Time-lapse fluorescence imaging and FRAP analyses demonstrate that Tau droplets mature significantly faster in the presence of Aβ40. This liquid-to-solid phase transition represents a key inflection point in the aggregation pathway, as more rigid condensates are often considered precursors to irreversible fibrillar assemblies. Aβ40 may facilitate this transition by stabilizing intermolecular contacts within the droplet and reducing molecular mobility, promoting the formation of aggregation-prone conformational states. Importantly, these Aβ40-induced changes in condensate dynamics appear to lower the energy barrier for nucleation, priming Tau for subsequent fibril formation. Such behavior aligns with existing models in which the phase maturation of condensates plays a critical role in the conversion of functional assemblies into pathological aggregates[20,24,25]. The observed co-condensation of Tau and Aβ40 further supports the notion that these heterotypic condensates act as nucleation sites for disease-associated aggregation. These findings are consistent with neuropathological observations showing Aβ and Tau co-localization in AD brain tissue and with the formation of NFTs, hallmark lesions that correlate with cognitive decline[9,10].

Our NMR results show that Aβ initially facilitates the solubilization of Tau into an NMR-detectable state. This effect occurs both in the presence and absence of macromolecular crowding, suggesting that the interaction is intrinsic. The NMR findings are further corroborated by mass photometry data, which show that Aβ promotes the presence of Tau monomers and low molecular oligomers in dilute solution. The consistency across these two orthogonal techniques supports a model in which Aβ transiently stabilizes soluble Tau species. Our data indicate that at early stages, Aβ may act to disperse Tau nanocluster into smaller, more soluble forms. Such solubilization could represent a priming mechanism, wherein monomeric Tau is temporarily stabilized but rendered more conformationally flexible or accessible for subsequent pathological transitions under aging-related conditions. One important implication is that Aβ's effect on Tau is likely stage-specific: neutral early on, but potentially harmful in later phases. The solubilized Tau, while initially innocuous, may serve as a substrate for subsequent co-condensation with Aβ, especially under conditions that favor LLPS. Therefore, this early interaction may set the stage for downstream pathological aggregation events.

Overall, our data collectively support a model in which Aβ facilitates the solubilization of Tau into an NMR detectable state. This early interaction is supported by NMR and mass photometry. However, upon incubation under crowding conditions, the nature of the interaction shifts. Tau and Aβ co-assemble into condensates and toxic oligomers, ultimately promoting a transition toward solid-like aggregates. This phase transition is supported by ThT fluorescence, LLPS assays, and cell viability. We propose that the initial solubilization and structural rearrangement of Tau by Aβ, while seemingly protective, may actually prime the system for subsequent pathological phase transitions. These later stages likely contribute to toxicity despite the transiently solubilized state observed at the beginning (Fig. 7).

Our MD simulations demonstrate that at near physiological temperature, Tau alone phase separates, whereas Aβ40 alone shows low

LLPS capacity. When mixed, Tau and Aβ40 co-phase separate into condensates that are denser than Tau alone and the saturation concentration of Tau is notably decreased by addition of Aβ40. Dynamic analyses further show that while Aβ40 remains relatively mobile, it reduces the dynamics of Tau molecules within the Tau-Aβ40 condensates, providing support to the experimentally observed reduction in FRAP recovery. Aβ40 stabilizes Tau condensates mainly by binding to the 4R repeat domains through hydrophobic and charged residues, thereby strengthening Tau-Tau electrostatic and hydrophobic contacts. These results provide mechanical insights into the experimental observation that Aβ40 promotes the phase separation and liquid-to-solid transition of Tau.

In our previous study, we demonstrated that Tau interacts with Aβ through two distinct modes[12]: weak interactions with soluble Aβ oligomers, which promote their partial disassembly into monomers, and higher-affinity interactions with Aβ fibrils, which inhibit further aggregation through a capping-like mechanism. In the present study, we observed a complementary phenomenon in which Aβ40 can act on dissolving Tau nanoclusters and remaining a more monomeric or smaller nanocluster state. Together, these results suggest that Tau–Aβ interactions are inherently bidirectional. Our data reveal a biphasic process: in the early stage, Tau and Aβ mutually solubilize each other's nascent assemblies through transient, weak interactions, leading to an initial increase in the monomeric forms of both proteins. However, over time, these solubilized Tau–Aβ complexes promote Tau phase separation and transition, suggesting that the initial solubilization effect is followed by a phase where heterotypic interactions facilitate rather than inhibit aggregate maturation. This temporal transition may reflect the evolution from dynamic, reversible interactions to more stable, aggregation-prone complexes.

Our findings reinforce the idea that AD pathology cannot be attributed solely to either Aβ or Tau but rather to their dynamic and bidirectional interactions. Our study provides insights into how Aβ40 modulates Tau phase separation and aggregation, highlighting the need for therapeutic approaches that target both proteins simultaneously. From a therapeutic perspective, strategies aimed at disrupting Aβ-Tau interactions could be particularly effective. For instance, small molecules or antibodies that prevent Aβ40 from being recruited into Tau condensates could mitigate the exacerbation of Tau pathology. Similarly, modulating LLPS through phase-separation inhibitors could prevent the formation of Tau-rich condensates that seed aggregation. Additionally, stabilizing Tau in its liquid-like phase while preventing its transition into a solid state could represent a strategy to delay disease progression[74]. Moreover, our results suggest that the phase-separation properties of Tau should be carefully considered when designing aggregation inhibitors. Traditional anti-aggregation compounds focus on disrupting fibril formation, but our findings indicate that intervening at the LLPS stage could be a more effective strategy. Given that phase separation precedes aggregation, early-stage interventions targeting this process could prevent the downstream formation of NFTs and Aβ plaques.

In conclusion, our study provides compelling evidence that Aβ40 modulates Tau LLPS, accelerates its phase transition, and enhances its fibrillation. These findings highlight the intricate relationship between Aβ and Tau in AD pathology and emphasize the need for therapeutic strategies that target their interactions. By elucidating the molecular mechanisms underlying Aβ-Tau interplay, our work contributes to a deeper understanding of AD and opens avenues for intervention aimed at mitigating disease progression.

## Methods
### Tau purification
The purification of recombinant full-length human Tau protein (Uniport P10636-8) was conducted following established protocols[75]. *E. coli* BL21 (DE3) cells were cultured in LB medium supplemented with ampicillin (100 mg/L). Protein expression was induced with 0.4 mM isopropyl β-D-1-thiogalactopyranoside (IPTG) at 0.6–0.8 of OD600, followed by incubation at 37 °C for 2 h. Cells were harvested by centrifugation, and the resulting pellets were stored at −20 °C prior to purification. For purification, Tau pellets were resuspended in 50 mM sodium phosphate buffer, 2.5 mM EDTA, pH 6.2. Following sonication, lysates were centrifuged at 5000 × *g* for 30 min to remove insoluble components. The supernatants were subjected to heat treatment at 75 °C for 15 min, followed by centrifugation at 25,000 × *g* for 20 min. The supernatant was loaded onto a Hi-Trap SP FF cation-exchange chromatography column (Cytiva) and eluted using a NaCl gradient. Protein was identified via SDS-PAGE and dialyzed against 20 mM ammonium bicarbonate buffer before lyophilization. Protein concentrations were determined by UV absorption at 274 nm, using extinction coefficients of 7600 M$^{-1}$ cm$^{-1}$ for Tau.

To label Tau with $^{15}$N isotopes, the *E. coli* BL21 (DE3) culture expressing Tau protein was grown in a M9 minimal medium with $^{15}$NH$_4$Cl and purified as described above.

Fluorescent labeling of Tau was performed with an Alexa Fluor™ 647 C2 Maleimide dye (Excitation maximum at 652 nm and emission maximum at 670 nm, Catalog number A20347, ThermoFisher Scientific, USA) following the manufactory protocol.

### Aβ40 sample preparation
Recombinant Aβ40 was purchased from AlexoTech, Umeå, Sweden (cat: AB-100-10). 1 mg of Aβ40 were dissolved to a concentration of 461 µM in 10 mM NaOH, and then sonicated in an ice-water bath for 1 min. The samples were filtered with a 0.22 µm centrifuge filter at 4 °C and aliquoted on ice and kept at −80 °C. TAMRA fluorescent labeled Aβ40 was purchased from Cayman Chemical (Excitation maximum at 552 nm and emission maximum at 578 nm, Catalog number: 27413).

### ThT fluorescence assay
To investigate amyloid fibrillation kinetics, 15 µM Tau was incubated with or without Aβ40 peptides in the presence of 50 µM Thioflavin T (ThT). All samples were prepared on ice, and 15 µL of each was transferred into a 384-well Black/Clear Bottom plate (cat: 242764, ThermoFisher) and sealed. Kinetics assays were performed using a PHERAstar FSX microplate reader (BMG LABTECH, Germany), recording fluorescence every 10 min with excitation and emission wavelengths of 430 nm and 480 nm, respectively. Assays were conducted in five replicates under shaking conditions (280 rpm) at 37 °C in 50 mM phosphate buffer (PB) with 150 mM NaCl (pH 7.4).

### LLPS
The LLPS experiments were conducted using a lipidic cubic phase (LCP) (Catalog number MD11-005U-20, Molecular Dimensions, UK) crystallization plate. All protein stock solutions were prepared on ice, and protein mixtures were pre-mixed at specific concentrations. Sample dispensing was performed using a multi-channel pipetting robot (Mosquito, SPT Labtech), where 200 nL of protein solution was mixed in a 1:1 ratio with buffer to achieve varying protein and PEG-8000 concentrations. Each condition was tested in at least three replicates. The plates were sealed, incubated at 20 °C, and imaged using a Rock Imager microscope (Formulatrix). Droplet size and area were quantified using Fiji software.

### Fluorescence imaging and FRAP analysis in vitro
Fluorophore-labeled LLPS experiments were conducted using a Stellaris microscope (Leica Microsystems, Germany). Droplet formation was achieved by mixing 5% labeled protein with 95% unlabeled protein, following the same sample preparation protocol as the unlabeled system. LCP plates were retrieved from the Rock Imager system and mounted onto the Stellaris microscope. Imaging was performed using a 647 nm laser line and a 63× oil-immersion lens.

Color crosstalk was tested and found to be negligible under our conditions.

Fluorescence recovery after photobleaching (FRAP) experiments were carried out on the same microscope in FRAP mode. For each droplet, a selected region of interest (ROI) was bleached using 60% laser intensity for 2 s. Post-bleach time-lapse images were acquired at a frame rate of 1 s for 60 frames and analyzed using the FRAP_profiler_v2 plugin developed by the Hardin lab in Fiji[76]. Fluorescence intensities were recorded for two ROIs: ROI1 (photo-bleached region) and ROI2 (unbleached region, used for background correction). The recovery time constant was determined by fitting the corrected fluorescence intensities to a single-exponential model.

## Turbidity measurements

The turbidity was measured in a PHERAstar FSX microplate reader (BMG LABTECH, Germany) in 384-well Black/Clear Bottom plates (cat: 242764, ThermoFisher) at room temperature. All samples were prepared on ice, and 30 µL of each sample was transferred to each well. The background absorbance was subtracted from the sample measurements, measured with 6 replicates and presented as average. The protein concentration and buffer condition were presented in the figure legend.

## NMR

The NMR experiments were performed on a 700 MHz Bruker spectrometer equipped with a triple resonance cryogenic probe. 100 µM $^{15}$N-Tau protein in 20 mM sodium phosphate (pH 6.3) alone or using a 1:1 Tau: Aβ40 molar ratio was used. Additional experiments were performed with 10% (w/vol) PEG8000 for LLPS conditions using 100 µM $^{15}$N-Tau. The experiments were performed at 283 K with 90/10 H$_2$O/D$_2$O. The spectra with and without Aβ40, both in aqueous solution and during LLPS conditions were compared by (1) calculating the relative intensities based on the amplitude intensities and (2) determination of the CSP. The CSP were calculated from equation below:

$$\Delta\delta = \left( \left( \left( \frac{\Delta\delta_N}{5} \right)^2 + \left( \Delta\delta_H \right)^2 \right) / 2 \right)^{1/2} \tag{1}$$

The cross-peak assignment of Tau (BMRB ID 50701) was achieved by a comparison to published work[77]. Data were processed and analyzed with software's Topspin v.4.2.0 (Bruker) and Poky[78] (University of Colorado Denver).

## Dot blot

Protein samples including Aβ40, Tau, and Tau-Aβ40 mixtures were prepared on ice. All samples were incubated at 37 °C for 2 h in 50 mM PB, 150 mM NaCl, pH 7.4, with or without 10% PEG8000. Subsequently, 3 µL of each sample was transferred onto a nitrocellulose membrane (Cat. No. 10600124; Amersham). The membrane was then blocked with 5% non-fat milk in TBS-T (20 mM Tris-HCl, 0.8% NaCl, 0.1% Tween 20, pH 7.4) for 1 h at room temperature, followed by overnight incubation at 4 °C with the primary antibody A11 (kindly provided by Prof. Rakez Kayed's group and used at a 1:1000 dilution). After three washes with TBS-T (10 min each), the membrane was incubated with a goat anti-rabbit IgG H&L (HRP) secondary antibody (Thermo Fisher Scientific, catalog no. 3246, 1:5000 dilution) for 1 h at room temperature. Following three additional washes with TBS-T, the membrane was developed using of the chemiluminescent substrate LumiGLO (Chemiluminescent Substrate Kit). The specificity of the primary antibodies used in this study has been previously validated, as described in the literature[59], and manufacturer datasheets.

## Cytotoxicity assays

SH-SY5Y human neuroblastoma cells (female) were obtained from Abcam (catalog number ab275475) and used for cytotoxicity assays. Cell line authentication was not independently performed in this study. The identity of the cell line was assured by the supplier. The cells were cultured in Dulbecco's Modified Eagle Medium (DMEM) supplemented with 10% fetal bovine serum (FBS) and 1% penicillin/streptomycin. The cells were maintained in a humidified incubator at 37 °C with 5% CO$_2$. For the experiment, cells were seeded into a 96-well plate at a density of 10,000 cells per well and allowed to adhere for 24 h. Aβ40, Tau, Tau: Aβ40 at ratios: 1:1 and 1:2 were prepared. All samples were pre-incubated at 37 °C for 2 h in 50 mM PB, 150 mM NaCl, pH 7.4, with or without 10% PEG8000. These pre-incubated samples were then added to the culture medium and incubated with the cells for 48 h. Each condition was tested in triplicate. Cytotoxicity was assessed using the CellTiter-Glo® Luminescent Cell Viability Assay Kit (Promega). For a 96-well plate, 100 µL of kit reagent was added per well. The plate was then shaken for 2 min and incubated at room temperature for 10 min. Luminescence was measured using a PHERAstar FSX microplate reader (BMG LABTECH). The intensity of the luminescent signal corresponded to cytotoxicity, with lower luminescence indicating greater cytotoxicity.

## Confocal images for cell uptake

SH-SY5Y cells were seeded at a density of 10,000 cells per well in 96-well plates. For visualization, Alexa Fluor 647–labeled Tau and TAMRA-labeled Aβ40 were included at 1% of the total protein concentration. Protein samples containing 15 µM Aβ40, 15 µM Tau, or a mixture of 15 µM Aβ40 and 15 µM Tau were pre-incubated at 37 °C for 2 h before being transferred to the cell culture medium. After 24 h of incubation, cells were washed three times with PBS (10 min each wash) at room temperature. Confocal images were acquired using a 20× objective. Red fluorescence signal indicates Tau protein (Alexa Fluor 647, excitation ~650 nm, emission ~668 nm), and green fluorescence singal indicates Aβ40 protein (TAMRA, excitation ~552 nm, emission ~578 nm).

## Mass photometry

Mass photometry was conducted using the OneMP mass photometer (Refeyn, UK) following a previous protocal[75]. High-precision coverslips culture well gaskets were cleaned following the manufacturer's protocol, involving three cycles of washing with isopropanol and Milli-Q water. For calibration, β-Amylase (monomer: 56 kDa, dimer: 112 kDa, tetramer: 224 kDa) and Thyroglobulin (monomer: 335 kDa, dimer: 670 kDa) served as standard proteins. Background subtraction was performed with buffer solution. Calibration was conducted using a laser for 120 s under optimized auto-exposure settings.

To monitor Aβ40 and Tau nanocluster formation over time, proteins were prepared in a 50 mM PB, 150 mM NaCl buffer (pH 7.4) to a total volume of 50 µL in a 1.5 mL Eppendorf tube. For mass photometry measurements, 18 µL of the buffer was first loaded onto the gaskets for autofocusing. Subsequently, 2 µL of sample was added inside. Measurements were performed with the laser for 60 s.

Data analysis was conducted using DiscoverMP software. Molecular weight estimations were based on the contrast values of individual landing events captured across multiple frames, generating a contrast-to-molecular weight (MW) calibration curve. Histograms of detected particles were fitted with Gaussian functions to determine the average MW of the samples. The detection limit for protein molecular weight was approximately 40 kDa, so individual Aβ40 monomers (~4 kDa) were not visible, while nanoclusters larger than 40 kDa were expected to be detectable, as previously reported[79].

## Coarse-grained molecular dynamics simulations

The full-length Tau and Aβ40 molecules were modeled as elastic chains of coarse-grained beads and each bead corresponds to a single

residue[44]. Sequential beads were connected by a harmonic bond with an equilibrium bond length of 0.38 nm and a spring constant of 1000 kJ/mol/nm². The nonbonded interactions in the model incorporated a Lenard-Jones term, utilizing the Hydropathy Scale (HPS) potential originally developed by the Mittal group[43] based on hydrophobicity scores calculated by Kapcha and Rossky[80].

$$\Phi_{nb}(r) = \begin{cases} 4\epsilon\left[\left(\frac{\sigma}{r}\right)^{12} - \left(\frac{\sigma}{r}\right)^6\right] + (1-\lambda)\epsilon & r \leq 2^{\frac{1}{6}}\sigma \\ 4\lambda\epsilon\left[\left(\frac{\sigma}{r}\right)^{12} - \left(\frac{\sigma}{r}\right)^6\right] & otherwise \end{cases} \quad (1)$$

The long-range electrostatic interactions were accounted for using a Coulomb term with Debye screening, with the relative permittivity denoted as $D$.

$$E_{ij}(r) = \frac{q_i q_j}{4\pi D r}\exp\left(-\frac{r}{\lambda_D}\right) \quad (2)$$

The Debye length $\lambda_D$ was set according to the Debye-Hückel formula, where $I$ corresponds to the ion strength which was set as 150 mM.

$$\lambda_D = \sqrt{\frac{\epsilon_r \epsilon_0 k_B T}{2e^2 I}} \quad (3)$$

To assess the LLPS capability of the Tau alone, Aβ40 alone and Tau-Aβ40 systems, we conducted direct phase-coexistence simulations. The initial configurations of Tau/ Aβ40 systems were generated by placing 80/160 chains of Tau/Aβ40 into a cuboid box with the centroids of the chains positioned at the grid points of a 4 × 4 × 5/ 5 × 5 × 7 evenly distributed grid. The initial configurations of Tau-Aβ40 systems were generated by splicing the above-mentioned Tau and Aβ40 simulation box. The configuration of each Tau or Aβ40 chain was generated via random walk, with the maximum distance from the centroid restricted to within 4 nm (for Tau) or 2 nm (for Aβ40). Each system was relaxed at a constant temperature of 150 K and a constant pressure of 1 bar using a Berendsen barostat, resulting in a compressed simulation box. Subsequently, the box was expanded in the z-direction by a factor of ten, creating a slab-like box. The temperature was then gradually increased to the target temperature at a rate of 0.2 K/ns in the NVT ensemble. The production run was conducted for 3.0 μs using a Langevin thermostat with a relaxation time of 5 ps and a time step of 10 fs. All simulations were carried out utilizing the HOOMD-Blue 2.9 package[81] on GPU cards. Trajectory analyses were carried out using our in-house-developed tools. The densities of the dense and dilute phases were obtained by fitting the density profiles to the following equation, where $z_0$ denotes the midpoint of the interface between the two phases, and $w$ represents the interfacial width.

$$\rho(z) = \frac{\rho_{dense} + \rho_{dilute}}{2} - \frac{\rho_{dense} - \rho_{dilute}}{2} \cdot \tanh\left[\frac{z - z_0}{w}\right] \quad (4)$$

We then determined the critical temperature by fitting the densities of the dense and dilute phases to the following critical equation, where $A$ is a universal fitting parameter and $\beta$ is the critical exponent, fixed at 0.365, corresponding to the universality class of the three-dimensional Heisenberg model.

$$\rho_{dense} - \rho_{dilute} = A(T - T_C)^{\beta} \quad (5)$$

The error bars for the densities shown in the phase diagram represent the 95% confidence intervals of the fitted density values for the two phases, while those for the critical temperature correspond to the 95% confidence intervals of the fitted critical temperature values. The cutoff for two residues forming a contact is calculated by 1.2*($\sigma_i + \sigma_j$)/2, in which $\sigma_i$ and $\sigma_j$ refer to the effective particle size defined as the distance at which the LJ potential equals zero[82,83]. The number of contacts between two protein molecules was defined as the number of intermolecular bead pairs within this cutoff distance. Reported contact numbers for Tau-Tau and Tau-Aβ pairs were normalized by the total number of Tau molecules in the system, while contact numbers for Aβ-Aβ pairs were normalized by the total number of Aβ molecules. To estimate the error bars for intermolecular and intramolecular contact numbers, we calculated the contact number for four equal segments of the final 1 μs of the trajectory. The error bars correspond to the minimum and maximum contact number values obtained from these four segments. The categorization of the 20 amino acids into hydrophobic, charged, polar, and other groups follows a previous study[50], with both hydrophobic and aromatic residues classified as hydrophobic for simplification.

### Quantification and statistical analysis
All statistical analysis was performed in GraphPad Prism or Origin. Data are presented as mean ± SEM (standard error of the mean) from at least three independent biological replicates, unless indicated otherwise. Statistical significance between experimental groups was analyzed either by two-tailed Student's t-test or one-way ANOVA followed by Bonferroni's multiple comparison.

### Reporting summary
Further information on research design is available in the Nature Portfolio Reporting Summary linked to this article.

## Data availability
The source data are provided in the Source Data file. The structure files for the initial and final states of all coarse-grained simulations are available on GitHub [https://github.com/ghwei-fudan/tau_abeta_phase_separation]. Source data are provided with this paper.

## Code availability
The codes generated in this study are available on GitHub [https://github.com/ghwei-fudan/tau_abeta_phase_separation].

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

## Acknowledgements

This work was supported financially by the China Scholarship Council (202004910346 to X.S.), the Swiss National Science Foundation (Grant No. 10002967 to J.L.), the National Key Research and Development Program of China (Grant No. 2023YFF1204402 to G.W.) and the National Natural Science Foundation of China (Grant No. 12374208 to G.W.). All MD simulations were performed using the GPU Cluster at Fudan University.

## Author contributions

X.S. and J.H.L. conceptualized the study. X.S., Y.T., X.W., G.P.C., R.S.H., C.M., and J.A.G. performed the experiment and developed the methodology. Y.T. and G.W. performed MD simulations, analyzed the data, and drafted the MD simulation part. X.W. performed the dot blot, mass photometry, and cell experiments, analyzed the data, and drafted the toxicity part. X.S., G.P.C., C.M., J.A.G., R.R. performed NMR data collection and data analysis. J.H.L. supervised all aspects of the study. The original manuscript draft was written by X.S., J.H.L., Y.T., G.W., and X.W., with review and editing by all authors.

## Competing interests

The authors declare no competing interests.
