## [Transparent Peer Review file · Nature Communications]

Alzheimer's A β Catalyzes Tau Phase Separation and Aggregation via Early Nanocluster Solubilization

Corresponding Author: Dr Jinghui Luo

Version 0:

Reviewer comments:

Reviewer #1

(Remarks to the Author)

Review attached.

Reviewer #2

(Remarks to the Author)

The paper is both interesting and timely. The observation of the acceleration of the LLPS of Tau in the presence of A β would be of great interest to those study Alzheimer's disease. The potential crosstalk of Tau/ A β in AD is nicely addressed here. There are a number of issues that might be modified to clarify the message in the paper.

Critics:

- 1) Why do the authors only use AB40 ? Why not AB42 ? It seems that these could have been carried out in parallel ?
- 2) A β is recruited into Tau condensates in this study. The authors state that A β and tau may be co-localised in the brain and cite a couple of papers. Are they proposing that this interaction is a switch to initiate Tau aggregation and propagation as observed in Alzheimer's disease or do they really expect A β to be found in most fibrils ? This message might be implicit in the paper but it is not explicit – Tau is involved in over 20 neurodegenerative diseases and it's cryo-EM structure has been solved for many of these diseases. Is A β 40 the switch to drive Tau to one of these forms ? It is a pity that the study could not have also been carried out using phospho-Tau, but that is a complicated thing to do.
- 3) In a previous paper, the group collaborated with another NMR team, that time using N-labelled A β and unlabeled Tau. The conclusion of that paper was that Tau inhibited A β aggregation by (i) monomerizing soluble oligomers into monomers with a weak affinity for Tau and (ii) binding to higher affinity A β fibrils to prevent further oligomerization by a kind of capping mechanism. Here they suggest that A β has the same monomerizing effect on small Tau nanoclusters. This is briefly touched on in the discussion. I think this message could be expanded upon to explain how you can have both of these effects and the role of the differing concentrations of each protein.
- 4) The conditions tested for LLPS of A β are very limited. LLPS of A β has been shown previously- but under acidic conditions. Do these conditions ever occur under physiological conditions ?
- 5) What about colour crosstalk of Alexa647 excited by TAMRA excitation ? Could the authors give a bit more information about those experiments to clarify how they were carried out.
- 6) The NMR data does seem to confirm the mass photometry data, however it gives very limited information on protein structure and interaction. The weak CSPs are unconvincing in terms of which residues really bind A β . The only vague information on structure is the Thioflavin data which is hardly conclusive. While the NMR data itself is not to be criticized, the lack of structural data is the weakest point of the paper. It would be very useful to have complementary structural data. Would it be possible to have Cryo-EM data of the aggregates ? Or failing that, some Circular Dichroism or Infra-Red data ?
- 7) From 216 – the authors talk About papers studying the effects on structure by polyanions. However the authors themselves give no structural information on their experiments other than a positive Thioflavin signal. As far as I am aware, the cryo-EM structure of Tau induced by mixing with 2 small beads is unknown. Have the authors any insight on the structure induced by their aggregation experiments ?
- 8) The authors carry out toxicity experiments on a cell line. Do the proteins enter the cells in the cell experiments ? Do they

remain together ? Have they had a look using their fluorophore labelled proteins ?

English errors:

78 biomolecule should be biomolecules

92 disease-related mutations Tau should be "disease-related tau mutation"

169/170 with or with should be "with or without"

187 remove "the" from we did the long term

241 isotopically labelled (labelled missing)

249 - A word or words are missing from this phrase.

Reviewer #3

(Remarks to the Author)

Version 1:

Reviewer comments:

Reviewer #1

(Remarks to the Author)

I would like to thank the authors for addressing the previous comments. There are a few points which need further clarification or justification.

From major comments:

Point 2. Fig. S3 (B): Tracking droplet diameters seems like a reasonable method to quantify droplet formation kinetics, assuming that key parameters are held constant (such as protein and crowder concentration). I assume that there is 10 μ M Tau and 10% PEG in both the cases. In this specific measurement, it appears that the slope for the diameter growth for Tau was considerably lower compared to Tau+AB. But after 4 hours, the slope of the Tau only case is considerably larger than the latter. How should one understand this difference?

Point 3. How were the errors in the critical point estimates determined? Also figure 4F still does not include error bars for the other points (dense and dilute phase measurement). Without these details, I am not convinced by the 0.1 to 0.3 K error for T_c .

Point 7. What is the justification for the 0.8nm cut off? Ideally, the cutoff should be based on residue-residue sigma values and not some arbitrary value. That is, a larger cutoff for pairs that have a larger sigma and smaller ones for pairs with lower sigma.

Form minor comments:

Point 5. The marker size in the legend could be further increased (The scatter points can remain as is).

Point 9. The reframed sentence is more or less same as the previous version and does not address the issue. The contact numbers for the different residue types are not that different and are possibly within error bars of each other. This leads me to an additional comment: There are no error bars on any of the bar plots in Fig. 5. I believe that the error bars are essential to support/deny the authors claims that polar and charged residues are the primary drivers of phase separation in their study. I suspect that the error bars would indicate that the contact numbers are within error limits of each other.

Reviewer #2

(Remarks to the Author)

The authors have addressed the concerns raised in a satisfactory manner.

Reviewer #3

(Remarks to the Author)

Version 2:

Reviewer comments:

Reviewer #1

(Remarks to the Author)

I would like to thank the authors for addressing my comments and answering my previous questions. After going through the responses, I still have some concerns about the contact analysis that has been performed:

Point 7. The primary concern about using a universal cutoff distance to determine contacts still holds. The three specific references that the authors have mentioned are not appropriate for the simulations/analysis performed in this study.

(a) Souza, P. C. T., et al. (2025). GōMartini 3: From large conformational changes in proteins to environmental bias corrections. *Nature Communications*, 16(1), 4051. This paper involves GoMartini simulations which are different from the CG HPS simulations, not just in terms of the resolution but also the underlying concept. In Go simulations, there is a template 'native' structure and the 'contacts' are defined relative to that structure. This is different from the contacts in the current study.

(b) Fuglebakk, E., Reuter, N., & Hinsen, K. (2013). Evaluation of protein elastic network models based on an analysis of collective motions. *Journal of Chemical Theory and Computation*, 9(12), 5618–5628. This paper uses elastic network models which are again different from the CG HPS simulations performed in the current study. Also, from a cursory look, the 0.8 nm cutoff that they mention is regarding the bonds in the model, not residue-residue contacts.

(c) Zhou, H., & Zhou, Y. (2002). Folding rate prediction using total contact distance. *Biophysical Journal*, 82(1), 458–463. This paper talks about "total contact distance" which is a combination of "contact order" and "long range order". This has nothing to do with the coarse-grained HPS simulations that have been performed in the current study.

In my opinion, the references stated are inaccurate given the coarse-grained simulations that have been performed and therefore the reasoning for the 0.8 nm cutoff does not hold true. Instead, the authors should use residue specific cut-offs as is the norm. As an example, please refer to this paper: "Feito, Alejandro, et al. "Capturing single-molecule properties does not ensure accurate prediction of biomolecular phase diagrams." *Molecular Physics* 122.21-22 (2024): e2425757." See Supplementary material section SVI, where a cut-off of 1.2 ij has been used.

Having said that, it would also be my recommendation to the authors to go through the references once more to make sure that they are accurate.

Point 9.

Fig. 5C, x-labels are missing. Since the contact analysis was erroneous, we cannot make any comment on the contact maps and therefore the subsequent analysis presented in Fig. 5.

Reviewer #3

(Remarks to the Author)

Version 3:

Reviewer comments:

Reviewer #1

(Remarks to the Author)

I would like to thank the authors for addressing my concerns regarding the usage of universal cutoff distance for the contact analysis. I am happy that the revised analysis strengthens the message of the analysis. After this revision, I am happy to recommend the manuscript for publication.

Reviewer #3

(Remarks to the Author)

Reviewer #1 (Remarks to the Author):

In this manuscript the authors have investigated the interplay of A β 40 peptides with Tau protein that drives the liquid-solid transitions of Tau protein droplets. The authors have employed various experimental techniques such as fluorescence microscopy, FRAP, turbidity measurements, NMR to qualify and quantify Tau aggregation phenotype in the presence of A β 40 peptides, and performed coarse-grained molecular dynamics (MD) simulations to understand the protein interactions at the residue level.

Key results: The authors claim that A β 40 promotes Tau phase separation and accelerates its phase transition to form amyloid like solid aggregates. Through MD simulations, they showed that A β 40 and Tau interact mainly through electrostatic and hydrophobic interactions. They also claim that in the absence of crowders, addition of A β 40 reduced cell toxicity, while in the presence of crowders (PEG), A β 40 increased cell toxicity. Based on the observations, the authors have proposed a mechanism where A β 40 can interact with Tau to keep it in its monomeric form. However, in the presence of crowders, A β 40 also promotes the phase separation and transition of tau proteins, leading to disease pathology.

Overall, the manuscript is well-written and presents some interesting results. However, there are several conceptual and technical points, which call into question the certain conclusions of the work.

Response: We sincerely thank the reviewer for their thorough assessment of our manuscript.

Major comments:

1. The phrase “A β 40 does not undergo LLPS” has little meaning from a thermodynamic standpoint. Since phase separation is a thermodynamic process, the conditions must be specified. Also the authors used the catGRANULE algorithm to predict whether A β 40 undergoes LLPS. conditions is this algorithm predicting the phase separation propensity for? Related to that, what does the score from this algorithm mean. Is there a maximum/minimum value one can expect? Is there a threshold score that makes something undergo LLPS (say under physiological conditions)? The details there are quite vague and it is difficult to appreciate what exactly is being predicted.

Response: We appreciate the reviewer’s insightful comments regarding the statement on A β 40 LLPS, as well as the need to clarify the interpretation of catGRANULE predictions.

Clarification of phrasing: We agree that the phrase “A β 40 does not undergo LLPS” is imprecise. What we meant is that under the conditions tested in our experiments (buffer: 50 mM PB, 150 mM NaCl pH 7.4, A β 40 concentration: 20, 50, 100 μ M, 10% and 15% PEG8000 as a crowding agent, with or without 12h incubation), A β 40 did not form condensates detectable by fluorescence microscopy or turbidity measurements. We have revised the text accordingly to specify the experimental conditions under which this observation holds true.

catGRANULE algorithm details: catGRANULE predicts the propensity of a protein sequence to form condensates by integrating sequence features such as intrinsic disorder and low-complexity regions (Bolognesi et al., 2016; Klus et al., 2014). The scores are relative values rather than absolute thermodynamic quantities. 1) There is no fixed “maximum/minimum” score, but higher scores indicate stronger predicted phase-separation propensity. 2) Importantly, catGRANULE does not model specific solution conditions (e.g., salt, crowding, pH), so the prediction should be interpreted as a sequence-intrinsic tendency, not as a definitive thermodynamic prediction under physiological conditions.

We have now specified in the Results that catGRANULE predictions provide a sequence-based propensity score and that experimental validation is required to confirm LLPS under specific conditions. The phrasing in the Results has been adjusted to:

“Overall, our experimental observations show that, unlike Tau, A β 40 does not form droplets under the tested conditions. This finding is consistent with catGRANULE predictions which indicate a much lower phase separation propensity of A β 40 compared to Tau, and is further supported by our coarse-grained simulations presented in the following sections.” (Page 4, line 139-143)

2. I find it difficult to see how the droplet area comparisons with and without A β 40 directly demonstrate accelerated kinetics of droplet formation. A β 40 alone produces droplets with a notably different topology than Tau alone (see Fig. 2A). Thus, the observed differences in droplet areas could largely reflect intrinsic differences in droplet morphology rather than kinetic effects. Consequently, additional evidence beyond droplet area measurements is required to substantiate the claim that A β 40 accelerates the kinetics of Tau droplet formation.

Response: We thank the reviewer for this insightful comment. We agree that differences in droplet area alone may not provide sufficient evidence to establish accelerated kinetics, since they could also reflect intrinsic morphological differences. To address this concern: 1) FRAP measurements (Fig 2D-E): We observed that Tau droplets formed with A β 40 exhibit reduced fluorescence recovery compared to Tau-only droplets, consistent with a more rapid liquid-to-solid transition. 2) time lapse imaging of Tau-A β 40 droplet formation: we observed that the maturation of Tau-A β 40 droplet is faster than Tau alone droplet and we quantify the diameter to establish the droplet formation kinetics in Fig S3A-B.

Together, these assays complement the microscopy data and support the interpretation of accelerated kinetics rather than purely morphological effects.

Figure S3. A β 40 accelerates Tau LLPS in vitro.

A, Representative time lapse images of phase separated droplets of Tau with or without A β 40 in the presence of 10% PEG. Scale bar in the images is 20 μ m.

B, Quantification of the diameter of droplets corresponding to A. The total number of droplets accounted is n=15 from 3 independent microscopy images.

3. From the coarse-grained MD simulations that authors claim that the dense-phase density in the presence of A β 40 is greater than when A β 40 is absent. However, the phase diagram in Figure 4F show practically identical dense-phase densities for both systems. Therefore, their results do not support this claim. Furthermore, there are no error bars shown in the figure. What is the error associated with the critical point estimate? It is likely that the 2 K difference

in T_c is within the uncertainty. Based on the results presented so far, it does not seem, from a thermodynamic standpoint, that the A β 40 is having a significant impact.

Response: We thank the reviewer for raising these insightful questions. The density profiles (Fig 4D-E) show that, near physiological temperature (290-320 K), the dense-phase densities of the Tau-A β 40 system are noticeably higher than those of the Tau-alone system. During the course of revision, to further support the impact of A β 40 on the phase separation capability of Tau, we introduced an additional metric, saturation concentration, which has been widely used to assess the phase separation propensity of protein systems¹⁻⁴. Our calculations reveal that, near physiological temperature (290-320 K), the saturation concentration of Tau molecules in the presence of A β 40 is markedly lower than in its absence, providing further evidence that A β 40 notably enhances the phase separation capability of Tau.

The phase diagrams (Fig 4F) show that the critical temperature of the Tau-A β 40 system is ~4.2 K higher than that of the Tau-alone system. This temperature difference is comparable to those reported in previous coarse-grained simulations assessing differences in phase separation capability between protein systems. For example, the D290V mutation, which experimentally reduces the phase separation capability of hnRNPA2, leads to a ~1% (~3 K) decrease in critical temperature in HPS simulations⁵. Removal of a C-terminal β -sheet-rich motif of α -synuclein results in a ~5 K decrease in critical temperature of phase separation in HPS simulations⁶. We acknowledge the absence of error bars in the original phase diagram. We have revised Figure 4F to include uncertainties and added an additional table (Table S1) to include the 95% confidence intervals of the critical temperature values. The uncertainties are 0.1 K for the Tau-alone system and 0.3 K for the Tau-A β 40 system, both of which are considerably smaller than the observed 4.2 K difference between the two systems. Taken together, these analyses provide further evidence that A β 40 notably enhances the phase separation propensity of Tau. The manuscript has been revised accordingly.

“We then calculated the saturation concentration of Tau to further quantify the impact of A β 40 on the phase separation capability of Tau. This metric has been widely used to evaluate the phase separation capabilities of protein systems¹⁻⁴. Our calculations show that the saturation concentration of Tau molecules in the presence of A β 40 is markedly lower than in the absence of A β 40 near physiological temperature (290-320 K), indicating that A β 40 enhances the phase separation propensity of Tau.” (Page 8, line 318-324)

“This difference in critical temperature (4.2 K) is comparable to those reported in previous coarse-grained simulations assessing differences in phase separation capability between protein systems⁵⁻⁶. For example, the D290V mutation, which experimentally reduces the phase separation capability of hnRNPA2, leads to a ~1% (~3 K) decrease in critical temperature in HPS simulations⁵. Removal of a C-terminal β -sheet-rich motif of α -synuclein results in a ~5 K decrease in critical temperature of phase separation in HPS simulations⁶.” (Page 8, line 313-318)

Method section: “The saturation concentration of Tau molecules is obtained by fitting the density profiles of Tau molecules to the Landau-Ginzburg free energy model, $\rho(z) = 0.5(\rho_{dense} - \rho_{dilute}) \cdot [\tanh(z - z_0/\xi)]$, where z_0 and ξ are fitting parameters that respectively correspond to the phase boundary and the correlation length.” (Page 37, line 969-972)

In addition, we have made the following changes to Figure 4 to improve readability. First, we added snapshots at 280, 290, and 360 K to Figures 4A-C to match the density profiles shown in Figures 4D-E. Second, because the original snapshots were taken from the final frame of each simulation, which was not always representative of the overall phase behavior, we replaced them with more representative frames. Accordingly, the figure caption has been

revised from “final simulation snapshots” to “representative simulation snapshots.” The main text has also been updated accordingly.

References:

1. Regy, Roshan Mammen, et al. "Sequence dependent phase separation of protein-polynucleotide mixtures elucidated using molecular simulations." *Nucleic acids research* 48.22 (2020): 12593-12603.
2. Martin, Erik W., et al. "A multi-step nucleation process determines the kinetics of prion-like domain phase separation." *Nature communications* 12.1 (2021): 4513.
3. Zhang, Yumeng, et al. "Toward accurate simulation of coupling between protein secondary structure and phase separation." *Journal of the American Chemical Society* 146.1 (2023): 342-357.
4. Kang, Wen Bin, et al. "Multi-scale molecular simulation of random peptide phase separation and its extended-to-compact structure transition driven by hydrophobic interactions." *Soft Matter* 19.41 (2023): 7944-7954.
5. Ryan, Veronica H., et al. "Mechanistic view of hnRNPA2 low-complexity domain structure, interactions, and phase separation altered by mutation and arginine methylation." *Molecular cell* 69.3 (2018): 465-479.
6. Tang, Yiming, et al. "Dissecting the Molecular Determinants of α - synuclein Phase Separation and Condensate Aging: The Pivotal Role of β - Sheet - Rich Motifs." *Advanced Science* (2025): e11545.

Figure 4. Phase separation propensity and phase behavior of Tau in the absence and presence of A β 40.

A-C, Representative simulation snapshots of Tau alone (A), A β 40 alone (B) and Tau-A β 40 (C) systems at six different temperatures.

D-E, Density profiles along the z-axis at eight temperature points for the Tau alone (D) and Tau-A β 40 (E) systems.

F, Phase diagrams of the Tau and Tau-A β 40 systems with critical temperatures obtained by fitting to critical equation.

G, The saturation concentration of Tau molecules of the Tau alone and Tau-A β 40 systems.

H-I, MSD of Tau molecules within Tau alone and Tau-A β 40 condensates (H) and that of A β 40 molecules within Tau-A β 40 condensate.

Table S1. Fitting of density profiles to critical function $y(t) = (\Delta\rho)^{1/\beta} = kT - kT_c$. The confidence bounds of each parameter are provided in the brackets.

System	k ($\times 10^6$ mg/mL/K)	T_c (K)
Tau alone	5.98 (5.93, 6.04)	347.30 (347.26, 347.35)
Tau-A β 40	5.85 (5.68, 5.99)	351.50 (351.31, 351.86)

4. Lines 366-368: “When Tau was co-incubated with A β 40, the aggregates displayed slightly increased toxicity compared to Tau alone (Fig 6B).” The difference there does not seem significant. They are both within the errors.

Response: We thank the reviewer for pointing this out. We agree that the difference in toxicity between Tau alone and Tau co-incubated with A β 40 is small and falls within experimental error. We have revised the text to avoid overinterpreting this trend.

“When Tau was co-incubated with A β 40, the resulting aggregates showed a trend toward slightly increased toxicity compared to Tau alone, although the difference was not statistically significant (Fig. 6B).” (Page 9, line 391-394)

5. Fig. 1F: At what time point of the droplet formations were the FRAP measurements performed? Something that seems contradictory to me is that the Tau droplet sizes increase rapidly in the presence of A β 40 due to coalescence (indicating liquid-like nature), while at the same time FRAP measurements are showing arrested dynamics. I would not have expected the two to be true at the same time. Were the FRAP measurements performed long aier (example, a day) the large droplets were already formed? Also, are similar FRAP measurements performed for A β 40 to confirm that all the dynamics are arrested?

Response: We thank the reviewer for this insightful comment. The FRAP measurements shown in Fig. 1F were performed at 1 day after droplet formation, when droplets had already reached their approximate equilibrium size. While rapid coalescence initially reflects the liquid-like nature of Tau droplets, it is known that these droplets can undergo aging, leading to slow or arrested internal dynamics over time (e.g., Wegmann et al., 2018). Therefore, the coalescence we observed does not contradict the FRAP results, which capture internal molecular mobility rather than droplet fusion.

FRAP measurements on A β 40 were also performed and are shown in Fig. S4. These data indicate that there is less fluorescence recovery for A β 40 FRAP behavior, confirming that the arrested dynamics in Tau+A β 40 droplets.

Figure S4. Representative FRAP images (up) and analysis (bottom) of 10 μM Tau droplet in the presence of 10 μM A β 40. The data represent the mean \pm SEM. (n=4). Buffer condition: 50 mM PB, 150 mM NaCl (pH 7.4), 10% PEG8000.

6. Fig. S9, Line 309 – 313: (1) In the MSD calculation for the Tau+A β 40, were the proteins treated separately? Tau is significantly longer than A β 40, which could lead to varying diffusion coefficient for the two species. It might be useful to separate the two to be accurate and maybe get some useful insight. (2) How exactly were the diffusion coefficient measured? Did the authors assume diffusive behavior and use the Stokes-Einstein relation? If so, that might be wrong since the MSD curves are clearly not linear and indicate sub-diffusive behavior (as expected). In that case one should be able to fit the MSD curve to $\langle r^2(\tau) \rangle = K\alpha\tau^\alpha$. Overall, the details on how the MSD calculations were done are lacking. What ensemble was used, how long were the simulations etc?

Response: We thank the reviewer for this insightful question. In our original manuscript, we calculated the MSD of protein molecules based on phase-coexistence simulations performed in slab-like simulation boxes. However, during the revision process, we learned that previous studies on MSD characterizations for biomolecular condensates are typically calculated from pure dense phases¹⁻², which were generated using cubic simulation boxes. We thus conducted additional simulations of the dense phase for both Tau alone and Tau-A β 40 systems, starting from preformed dense phases taken from the final frame of the original phase-coexistence simulations (Figure S8). We then recalculated the MSD curves using the last 1 μs of each simulation trajectory.

For the Tau-alone system, we calculated the MSD curve for all Tau molecules, whereas for the Tau-A β 40 system, we separately calculated the MSD curves for Tau and A β 40 molecules. Our results indicate that the MSD values for Tau molecules in the Tau-A β 40 system are smaller than those in the Tau-alone system (Figure 4H), suggesting that the presence of A β 40 reduces the fluidity of Tau molecules within the condensate. Additionally, the MSD of A β 40 in the Tau-A β 40 system is significantly larger than that of Tau (Figure 4H-I), indicating higher dynamics for A β 40 molecules. Interestingly, we observed that the MSD curves for Tau in both systems are not linear (Figure 4H), while the MSD curve for A β 40 is linear (Figure 4I) with respect to lag time. This suggests that Tau molecules exhibit sub-diffusive behavior, whereas A β 40 molecules follow diffusive behavior. We thus revise the calculation of diffusion coefficient for Tau and A β 40 molecules by fitting to power-law diffusion equation ($MSD = K_\alpha\tau^\alpha$) and Stokes-Einstein relation ($MSD = 6D\tau$), respectively. To highlight the importance of the MSD analysis, we have moved the corresponding figures from the Supporting Information to the main text

(Figure 4H-I). In addition, we have revised the manuscript to provide details on the MSD calculations.

“To assess the dynamics of the condensates, we followed previous studies¹⁻² and extracted the dense phases from the final state of the phase-coexistence simulations and placed them into cubic simulation boxes (Fig S8). The dense phases were then simulated for 5 μ s at 310 K in the NVT ensemble. Using the last 1 μ s of each trajectory, we calculated the mean square displacement (MSD) for Tau molecules in the Tau-alone and Tau-A β 40 systems, as well as the MSD for A β 40 molecules in the Tau-A β 40 system. The MSD values for Tau molecules display a non-linear behavior with lag time, suggesting sub-diffusive behaviors (Fig 4H). Fitting MSD curves to power-law diffusion equation $MSD = K_{\alpha}\tau^{\alpha}$ yields a smaller diffusion exponent for Tau molecules within the Tau-A β 40 condensate (0.494 ± 0.001) compared to those within Tau-alone condensate (0.500 ± 0.001), indicating that the presence of A β 40 reduces the fluidity of Tau molecules within the condensate. Furthermore, the MSD of A β 40 in the Tau-A β 40 system is significantly larger than that of Tau and follows a diffusive behavior characterized by the Stokes-Einstein relation ($MSD = 6D\tau$) with a diffusive coefficient (D) of 1.696 ± 0.004 nm²/ns (Fig 4I). These observations suggest that A β 40 molecules remain highly dynamic within the condensate while reducing the dynamics of Tau molecules, thereby providing further support for the A β 40-induced acceleration of Tau phase transition observed in our experiments.” (Page 8, line 324-341)

Figure S8. Snapshots at three representative time points from simulations of a preformed Tau condensate (A) and a Tau-A β 40 condensate (B) in a cubic simulation box.

Figure 4. H-I, MSD of Tau molecules within Tau alone and Tau-A β 40 condensates (H) and that of A β 40 molecules within Tau-A β 40 condensate.

References:

1. Sundaravadivelu Devarajan, Dinesh, et al. "Sequence-dependent material properties of biomolecular condensates and their relation to dilute phase conformations." *Nature Communications* 15.1 (2024): 1912.
2. Galvanetto, Nicola, et al. "Material properties of biomolecular condensates emerge from nanoscale dynamics." *Proceedings of the National Academy of Sciences* 122.23 (2025): e2424135122.

7. Line 317-318: Provide more information as to how the interaction analysis was performed. Was a distance threshold used? If so, what was it? Also, were the contact maps normalized in Fig. 5? Between residues 200-300 the contact maps have some contact values almost equal to 1. Does this mean that those residue pairs were in contact most of the time?

Response: We have added a paragraph in the Methods section to describe the interaction analysis details, including the distance threshold and normalization methods.

"Interaction analyses were conducted using a bead-to-bead distance cutoff of 0.8 nm. The number of contacts between two molecules was defined as the number of intermolecular bead pairs within this cutoff distance. Reported contact numbers for Tau-Tau and Tau-A β pairs were normalized by the total number of Tau molecules in the system, while contact numbers for A β -A β pairs were normalized by the total number of A β molecules." (Page 37, line 972-977)

We would like to point out that the color bars in Figures 5B and 5D range from 0 to 0.86, rather than from 0 to 1. Therefore, the contact probabilities of all residue pairs are not nearly equal to 1, as might be mistakenly inferred.

Figure 5.

B, Inter-molecular contact numbers between each pair of Tau residues in the Tau system. The accumulated contribution of each Tau residue to inter-molecular contacts is shown on the right. **D**, Inter-molecular contact numbers between each Tau residue and each Tau/A β 40 residue in the Tau-A β 40 system. The accumulated contribution of each Tau residue to inter-molecular contacts is shown on the right.

Minor comments:

1. Figure 6 B/E or C/F: Invert either the intensity or the cell viability so that both quantities vary in the same direction. This will make the comparison of B and C and E and F more straightforward.

Response: We thank the reviewer for this helpful suggestion. In Fig. 6C/F, we have now inverted cell viability to cell toxicity, which makes the comparison more straightforward.

“Consistently, in SH-SY5Y cells, Aβ40-Tau condensates exhibited higher toxicity compared to Tau alone (Fig 6F). Due to the high toxicity of the co-condensates, a slight increase in cell viability was observed as the Tau-Aβ40 ratio increased.” (Page 10, lines 414-417)

Figure legend updated in Figure 6: “SH-SY5Y cell toxicity under various conditions, including different concentrations of Aβ40, Tau, Tau: Aβ40 at 1:1 and 1:2 molar ratios in the presence of 10% PEG8000. All samples were pre-incubated at 37 °C for 2 hours, prior to a 48-hour incubation with cells. Buffer: 50 mM PB, 150 mM NaCl, pH 7.4. Statistical significance was determined by t-test (**p < 0.01, ***p < 0.001, ****p < 0.0001)” (Page 21, lines 681-685)

2. Lines 155-156: Which figures correspond to the brightfield microscopy images that is being alluded to?

Response: We thank the reviewer for pointing it out. The brightfield microscopy images correspond to Figure S2. We have revised sentence to ensure that readers can directly connect the description to the corresponding figure.

“These results were corroborated by brightfield microscopy images (Fig S2), which showed increased droplet formation in the presence of A β 40.” (Page 4, line 158-159)

3. Fig. 1A: Indicate the sequence of the Tau protein that has been used (and the uniport id). Is it P10636-8?

Response: We thank the reviewer for this point. We used the full-length human Tau protein that uniport id is P10636-8. We have now indicated Uniport ID in the figure legend of Fig. 1A and in the Methods section for clarity.

4. Line 169-170: There is a typo. Instead of “with or with” it should be “with or without”

Response: We have corrected the phrase to “with or without” in the revised manuscript.

5. Fig. 2H: The legend markers should be made larger.

Response: We have now increased the size of the legend markers in **Fig. 2H** in the revised figure to improve readability.

6. Line 249: Sentence reads odd. Missing word here “formed relatively nanocluster”?

Response: Thanks. We have rewritten “A β 40 at both 5 μ M and 25 μ M formed small nanoclusters with predominant mass distributions at ~90 kDa and ~84 kDa, respectively” in the revised manuscript.

7. Line 274: Typo. Instead of “imply that enhancement A β 40–Tau interactions”, it should be “imply enhancement of A β 40–Tau interactions.”

Response: Thanks. We have corrected it in the revised manuscript.

8. Fig. 5C: It would be helpful if you could indicate in the caption what the letters ‘H’, ‘P’, ‘C’, ‘O’ represent.

Response: The figure caption has been revised accordingly.

“C, Accumulated contact numbers of each Tau residue type with all residues, i.e., polar (P), charged (C), hydrophobic (H), and other (O) residues, in the Tau alone system.”

9. Line 333 – 335: “Our calculations indicated that charged residues were the primary drivers of Tau phase separation, followed by polar residues, with hydrophobic residues also making nonnegligible contributions (Fig 5C).” The way this sentence is written, it seems that the accumulated contact numbers for the electrostatic interactions are much larger than the other ones (polar and hydrophobic). However, from the figure the polar contacts are almost equal to the electrostatic ones and closely followed by the hydrophobic interactions. The sentence could perhaps be framed in a different manner.

Response: The sentence has been revised accordingly.

“Our calculations indicated that charged and polar residues were the primary drivers of Tau phase separation, with hydrophobic residues also making nonnegligible contributions (Fig 5C).”

Reviewer #2 (Remarks to the Author):

The paper is both interesting and timely. The observation of the acceleration of the LLPS of Tau in the presence of A β would be of great interest to those study Alzheimer's disease. The potential crosstalk of Tau/ A β in AD is nicely addressed here. There are a number of issues that might be modified to clarify the message in the paper.

Response: We sincerely thank the reviewer for their positive evaluation of our work.

Critics:

1. Why do the authors only use A β 40? Why not A β 42? It seems that these could have been carried out in parallel?

Response: We thank the reviewer for raising this important point. In this study, we focused on A β 40 for two main reasons: **1) Experimental tractability:** A β 42 is substantially more aggregation prone than A β 40. This presents technical challenges in maintaining monomeric or low molecular weight oligomeric A β 42 long enough to systematically study its effects on Tau phase separation across the same timescales and conditions. For this reason, we prioritized A β 40 to establish a clear experimental framework before extending the approach to A β 42. **2) Physiological abundance:** A β 40 is the most abundant isoform of A β in the human brain and cerebrospinal fluid, accounting for ~80–90% of total A β peptides, whereas A β 42 constitutes ~5–10%. Thus, we considered A β 40 to be a highly relevant A β variant starting point for studying Tau–A β crosstalk under near-physiological conditions.

2. A β is recruited into Tau condensates in this study. The authors state that A β and tau may be co-localized in the brain and cite a couple of papers. Are they proposing that this interaction is a switch to initiate Tau aggregation and propagation as observed in Alzheimer's disease or do they really expect A β to be found in most fibrils? This message might be implicit in the paper but it is not explicit – Tau is involved in over 20 neurodegenerative diseases and its cryo-EM structure has been solved for many of these diseases. Is A β 40 the switch to drive Tau to one of these forms? It is a pity that the study could not have also been carried out using phospho-Tau, but that is a complicated thing to do.

Response: We thank the reviewer for raising this important conceptual point. Our intention is not to propose that A β 40 is universally incorporated into Tau fibrils or that it serves as a singular “switch” to drive Tau into a specific fibril polymorph. Instead, our data suggest that A β 40 can modulate the phase behavior of Tau, lowering the barrier for Tau to undergo LLPS and promoting its transition toward solid-like aggregates under crowding conditions. This supports a model where A β 40 acts as a co-factor that facilitates Tau condensation and aggregation, rather than being an obligatory structural component of Tau fibrils.

Based on current research in this field, A β aggregation is thought to occur at early stages of the disease, and ultimately gives way to the formation of tau tangles (Tripathi T, et al. Biochemistry. 2020). The evidence doesn't strongly support that A β is found as a major component in most mature tau fibrils. The current evidence more strongly supports A β as playing a critical role in dissolving tau aggregation rather than being a persistent component of mature tau aggregates (Wallin C, et al. JACS. 2018). The co-localization observed in brain tissue may reflect this early interaction phase rather than stable co-incorporation into fibrils.

We agree with the reviewer that Tau is implicated in more than 20 neurodegenerative diseases, each with distinct fibril polymorphs revealed by cryo-EM. In the context of Alzheimer's disease, A β and Tau pathology are closely linked and co-localization has been observed. A β 40 may act as a switch that drives Tau into one or multiple fibrillar forms, as observed in different diseases. However, further cryo-EM studies will be necessary to validate these effects on Tau fibril polymorphs, which are beyond the scope of the present study. In the present work, our

data support an initial phase in which A β first solubilizes Tau and subsequently promotes its liquid–solid transition under crowding conditions.

Regarding phosphorylated Tau, we fully agree with the reviewer that this is a highly relevant and interesting direction, given the central role of phosphorylation in Tau pathology. Due to technical limitations and the complexity of producing site-specifically phosphorylated Tau, we restricted this study to unmodified Tau although phosphorylated Tau is more closely mimic the pathological state.

3. In a previous paper, the group collaborated with another NMR team, that time using N-labelled A β and unlabeled Tau. The conclusion of that paper was that Tau inhibited A β aggregation by (i) monomerizing soluble oligomers into monomers with a weak affinity for Tau and (ii) binding to higher affinity A β fibrils to prevent further oligomerization by a kind of capping mechanism. Here they suggest that A β has the same monomerizing effect on small Tau nanoclusters. This is briefly touched on in the discussion. I think this message could be expanded upon to explain how you can have both of these effects and the role of the differing concentrations of each protein.

Response: We thank the reviewer for this thoughtful comment and for drawing attention to our previous work. Indeed, in our earlier NMR study, we reported that Tau interacts with A β through two distinct modes: (i) weak interactions with soluble A β oligomers that promote their partial disassembly into monomers, and (ii) higher-affinity interactions with A β fibrils that limit further elongation through a capping-like mechanism. In the present study, we observed a complementary phenomenon, whereby A β_{40} acts on small, dissolving Tau nanoclusters to maintain a more monomeric state. Together, these findings suggest that Tau–A β interactions are inherently bidirectional and context-dependent. Our data indicate a biphasic process: at early stages, transient and weak heterotypic interactions between Tau and A β promote mutual solubilization of nascent assemblies, leading to an initial increase in monomeric species for both proteins. Over time, however, these solubilized Tau–A β complexes can evolve toward more stable, aggregation-prone states, facilitating Tau phase separation and transition. This temporal shift likely reflects a continuum from dynamic, reversible contacts to more persistent, aggregation-favoring interactions, thus reconciling the dual “monomerizing” and “stabilizing” effects observed across studies. We have now expanded the Discussion to explicitly link these findings.

“In our previous study, we demonstrated that Tau interacts with A β through two distinct modes¹²: weak interactions with soluble A β oligomers, which promote their partial disassembly into monomers, and higher-affinity interactions with A β fibrils, which inhibit further aggregation through a capping-like mechanism. In the present study, we observed a complementary phenomenon in which A β_{40} can act on dissolving small Tau nanoclusters and remaining a more monomeric state. Together, these results suggest that Tau–A β interactions are inherently bidirectional. Our data reveal a biphasic process: in the early stage, Tau and A β mutually solubilize each other’s nascent assemblies through transient, weak interactions, leading to an initial increase in monomeric of both proteins. However, over time, these solubilized Tau–A β complexes promote Tau phase separation and transition, suggesting that the initial solubilization effect is followed by a phase where heterotypic interactions facilitate rather than inhibit aggregate maturation. This temporal transition may reflect the evolution from dynamic, reversible interactions to more stable, aggregation-prone complexes.” (page 13, line 529-542)

4. The conditions tested for LLPS of A β are very limited. LLPS of A β has been shown previously- but under acidic conditions. Do these conditions ever occur under physiological conditions?

Response: We thank the reviewer for highlighting this important point. Indeed, prior studies have reported that A β peptides can undergo LLPS under acidic conditions, which differs from

the near-neutral physiological pH of the cellular environment. Our study focused on **physiological buffer conditions (pH ~7.4)**, where we did not observe droplet formation of A β 40 alone.

5. What about colour crosstalk of Alexa647 excited by TAMRA excitation? Could the authors give a bit more information about those experiments to clarify how they were carried out.

Response: We thank the reviewer for this helpful comment. We carefully considered the possibility of spectral crosstalk between TAMRA and Alexa647 in our microscopy experiments. To address this: **1)** Control measurements were performed with samples containing only Alexa647-labeled Tau and excited with the TAMRA excitation wavelength. These controls showed negligible signal, confirming that direct excitation of Alexa647 under TAMRA illumination is minimal. **2)** Sequential imaging was used to acquire TAMRA and Alexa647 channels separately, further minimizing potential bleed-through.

As shown in our microscopy setup, TAMRA and Alexa647 were excited and detected in fully separated channels. TAMRA was excited with a 561 nm laser and detected between 570–630 nm, while Alexa647 was excited with a 638 nm laser and detected above 650 nm. The two detection windows were separated by a spectral gap (630–650 nm) to avoid bleed-through.

We have now clarified these experimental details in the Methods section and added a statement that **color crosstalk was tested and found to be negligible under our conditions.**

6. The NMR data does seem to confirm the mass photometry data, however it gives very limited information on protein structure and interaction. The weak CSPs are unconvincing in terms of which residues really bind A β . The only vague information on structure is the Thioflavin data which is hardly conclusive. While the NMR data itself is not to be criticized, the lack of structural data is the weakest point of the paper. It would be very useful to have complementary structural data. Would it be possible to have Cryo-EM data of the aggregates? Or failing that, some Circular Dichroism or Infra-Red data?

Response: We thank the reviewer for this valuable comment. We agree that additional structural information would strengthen our conclusions. To address this, we have now performed Circular Dichroism (CD) spectroscopy on Tau in the presence and absence of A β 40. As we showed previously (Wallin C, et al. JACS. 2018), the results indicate that the mixed samples don't show a significant increase in negative ellipticity around 200 nm compared to

the added values of Tau alone and A β alone. This suggests that CD may not be sufficiently sensitive to detect the formation of soluble nanoclusters.

Furthermore, distinguishing between A β and Tau oligomers in the mixture is challenging by AFM and cryo-EM, as demonstrated in our previous study (Wallin C, et al. JACS. 2018). To assess the overall structural impact, we employed mass photometry (Fig. S6), which reveals that A β influences Tau oligomer formation.

Figure S7. CD spectroscopy measurements of Tau, A β , and their mixture. Far-UV CD spectra comparing 10 μ M Tau alone (red), 10 μ M A β alone (blue), Tau alone + A β alone signal (magenta) and the Tau-A β mixture (black). CD measurements were performed in the far-UV region (190-260 nm) and data are averaged from 3 triplicates and expressed as circular dichroism signal (mdeg). Buffer condition: 50 mM PB, 150 mM NaCl, pH 7.4.

“However, these structural and global conformational changes were not detectable by CD spectroscopy (Fig. S7).” (Page 7, line 266-267)

7. From 216 – the authors talk About papers studying the effects on structure by polyanions. However the authors themselves give no structural information on their experiments other than a positive Thioflavin signal. As far as I am aware, the cryo-EM structure of Tau induced by mixing with 2 small beads is unknown. Have the authors any insight on the structure induced by their aggregation experiments?

Response: We thank the reviewer for this insightful comment. We agree that high-resolution structural characterization of the Tau fibrils formed under our experimental conditions would provide valuable information. However, this question is beyond the scope of the present study, which was focused on the interplay between A β 40 and Tau during phase separation and aggregation.

The cryo-EM structure of Tau fibrils induced by glass beads has not yet been resolved, and we therefore cannot make direct structural assignments for the aggregates observed here. We fully agree that this is an important and interesting direction, and we are currently pursuing such structural studies as part of our next project.

8. The authors carry out toxicity experiments on a cell line. Do the proteins enter the cells in the cell experiments? Do they remain together? Have they had a look using their fluorophore labelled proteins?

Response: We thank the reviewer for this insightful question. In additional imaging experiments using fluorophore-labeled proteins, we observed that both A β 40 and Tau can enter cell, as shown in Fig. S14 A. However, because overall cell condition declined after 24h of protein incubation and imaging through plastic 96-well plates limited resolution, the Tau signal was faint, making it difficult to assess Tau alone and Tau/A β 40 colocalization in cell. Then we only calculated the total A β 40 fluorescence of each image in the A β 40 alone and A β 40+Tau systems. As shown in Fig. S14B, co-incubation with Tau increased intracellular A β 40 signal, which parallels the higher toxicity observed for the mixture relative to A β 40 alone (Fig. 6C). We hypothesize that A β 40 renders Tau more soluble, promoting Tau to associate with the cell membrane to form oligomers. These oligomers then insert into the lipid bilayer, increase membrane permeability, and in turn facilitate A β 40 uptake. This is consistent with prior reports that A β and Tau can damage cellular membranes and promote internalization. (Fernandez-Perez EJ, et al. *Curr Pharm Des.* 2016; Flach K, et al. *J Biol Chem.* 2012; Jones EM, et al. *Biochemistry.* 2012).

“Then we used fluorophore-labeled proteins to check if these proteins go into cells. We found that co-incubation with Tau increased intracellular A β 40 signal (Fig.S14A-B), which parallels the higher toxicity observed for the mixture relative to A β 40 alone. Both A β and Tau can damage cellular membranes and promote internalization and A β 40 renders Tau more soluble, promoting Tau to associate with the cell membrane to form oligomers. These oligomers then insert into the lipid bilayer, increase membrane permeability, and in turn facilitate A β 40 uptake. Further studies are required to validate this mechanism.” (Page 10, line399-405)

Figure S14. (A) Representative confocal images of SH-SY5Y cells after 24 h incubation with 15 μ M A β 40, 15 μ M Tau, or a mixture of 15 μ M A β 40 and 15 μ M Tau. A β 40 was detected in the TAMRA channel (shown in green) and Tau in the Alexa 647 channel (shown in red). Left column: transmitted light/fluorescence overlay; middle: TAMRA channel; right: Alexa 647 channel. Scale bar, 20 μ m. (B) Quantification of total A β 40 fluorescence intensity in the A β 40 alone and A β 40+Tau groups shown in (A).

Method section: “SH-SY5Y cells were seeded at a density of 10,000 cells per well in 96-well plates. For visualization, Alexa Fluor 647–labeled Tau and TAMRA-labeled A β 40 were included at 1% of the total protein concentration. Protein samples containing 15 μ M A β 40, 15 μ M Tau, or a mixture of 15 μ M A β 40 and 15 μ M Tau were pre-incubated at 37 °C for 2 h before being transferred to the cell culture medium. After 24 h of incubation, cells were washed three times with PBS (10 min each wash) at room temperature. Confocal images were acquired using a 20 \times objective. Red fluorescence signal indicates Tau protein, and green fluorescence signal indicates A β 40 protein. Scale bar: 20 μ m”. (Page 35, line905-912)

English errors:

78 biomolecule should be biomolecules

Response: Corrected it in the revised manuscript (page 2, line 78).

92 disease-related mutations Tau should be “disease-related tau mutation”

Response: Corrected it in the revised manuscript (page 3, line 92-93).

169/170 with or with should be “with or without”

Response: Rephrased to “with or without” in the revised manuscript (page 4, line 169-170).

187 remove “the” from we did the long term

Response: Removed it in the revised manuscript.

241 isotopically labelled (labelled missing)

Response: Added “labelled” in the revised manuscript (page 6, line 241).

249 - A word or words are missing from this phrase.

Response: We have rewritten “A β 40 at both 5 μ M and 25 μ M formed small nanoclusters with predominant mass distributions at ~90 kDa and ~84 kDa” in the revised manuscript (page 6, line 249).

Reviewer #3 (Remarks to the Author):

Response: We thank the reviewer and the co-reviewer for their careful evaluation of our manuscript.

Reviewer #1 (Remarks to the Author):

I would like to thank the authors for addressing the previous comments. There are a few points which need further clarification or justification.

From major comments:

Point 2. Fig. S3 (B): Tracking droplet diameters seems like a reasonable method to quantify droplet formation kinetics, assuming that key parameters are held constant (such as protein and crowder concentration). I assume that there is 10 μM Tau and 10% PEG in both the cases. In this specific measurement, it appears that the slope for the diameter growth for Tau was considerably lower compared to Tau+A β . But after 4 hours, the slope of the Tau only case is considerably larger than the latter. How should one understand this difference?

Response: We thank the reviewer for this insightful comment. As shown in Fig. S3, the kinetics of droplet growth differ between Tau and Tau+A β 40. During the early stage, Tau+A β 40 droplets form and grow more rapidly, suggesting that A β facilitates condensate formation and initial growth. However, after 4 h, the growth rate of Tau-only droplets becomes higher, resulting in larger droplets at later time points.

This crossover likely reflects a shift in the dominant mechanism of droplet maturation. The early acceleration observed in Tau+A β 40 samples result from rapid recruitment of Tau into numerous small droplets, as supported by microscopy imaging (Fig. S3) and turbidity measurements (Fig. 1C). Further A β 40 promotes liquid–solid phase transition, as indicated by FRAP analyses (Fig. 1F, Fig. 2D–F, and Fig. S4), and enhance fibril formation, consistent with ThT kinetics (Fig. 2H). Together, these effects lead to depletion of soluble Tau monomers and reduced droplet mobility, thereby slowing further growth after 4 h. In contrast, Tau-only samples display slower condensates formation with more droplet mobility at early stage and undergo more extensive coalescence and Ostwald ripening, resulting in continued droplet enlargement at later time points. We have added the text below to our manuscript (in the end of Page 4)

“The observed crossover in droplet growth kinetics (Fig S3B) where Tau+A β 40 droplets initially form and expand faster than Tau-only droplets, but later exhibit slower growth. This reflects an early A β 40-driven enhancement of condensate nucleation and maturation followed by reduced droplet mobility and Tau monomer depletion due to A β -induced liquid–solid transition and fibril formation, whereas Tau-only samples continue to enlarge through prolonged coalescence and Ostwald ripening.”

Point 3. How were the errors in the critical point estimates determined? Also figure 4F still does not include error bars for the other points (dense and dilute phase measurement). Without these details, I am not convinced by the 0.1 to 0.3 K error for T_c.

Response: We apologize for the insufficient details for our density calculations. We have revised the *Methods* section to include a detailed description of how the following quantities were determined, including the dense and dilute phase densities, critical temperatures, and their associated error bars. In addition, Figure 4F has been updated to display the corresponding error bars.

“The densities of the dense and dilute phases were obtained by fitting the density profiles to the following equation, where z_0 denotes the midpoint of the interface between the two phases, and w represents the interfacial width.

$$\rho(z) = \frac{\rho_{dense} + \rho_{dilute}}{2} - \frac{\rho_{dense} - \rho_{dilute}}{2} \cdot \tanh\left[\frac{z - z_0}{w}\right]$$

We then determined the critical temperature by fitting the densities of the dense and dilute phases to the following critical equation, where A is a universal fitting parameter and β is the critical exponent, fixed at 0.365, corresponding to the universality class of the three-dimensional Heisenberg model.

$$\rho_{dense} - \rho_{dilute} = A(T - T_c)^\beta$$

The error bars for the densities shown in the phase diagram represent the 95% confidence intervals of the fitted density values for the two phases, while those for the critical temperature correspond to the 95% confidence intervals of the fitted critical temperature values.”

Figure 4. Phase separation propensity and phase behavior of Tau in the absence and presence of A β 40.

A-C, Representative simulation snapshots of Tau alone (A), A β 40 alone (B) and Tau-A β 40 (C) systems at six different temperatures.

D-E, Density profiles along the z-axis at eight temperature points for the Tau alone (D) and Tau-A β 40 (E) systems.

F, Phase diagrams of the Tau and Tau-A β 40 systems with critical temperatures obtained by fitting to critical equation.

G, The saturation concentration of Tau molecules of Tau alone and Tau-A β 40 systems.

H-I, MSD of Tau molecules within Tau alone and Tau-A β 40 condensates (H) and that of A β 40 molecules within Tau-A β 40 condensate.

Point 7. What is the justification for the 0.8nm cut off? Ideally, the cutoff should be based on residue-residue sigma values and not some arbitrary value. That is, a larger cutoff for pairs that have a larger sigma and smaller ones for pairs with lower sigma.

Response: We use a universal cutoff of 0.8 nm for calculating both intra- and intermolecular interactions. This choice is consistent with previous computational studies of protein folding, protein-membrane, and intra-protein interactions using single-bead-per-residue coarse-grained (CG) protein force fields [1-3]. These studies have shown that a cutoff of 0.8 nm reliably captures the effective residue-residue interaction strength in such CG protein models, and that the folding rates are largely insensitive to cutoff values up to 1.0 nm [3]. Therefore, we adopted a cutoff of 0.8 nm in our study. We have added a discussion in the revised Methods section.

“Interaction analyses were conducted using a bead-to-bead distance cutoff of 0.8 nm. This choice is consistent with previous computational studies of protein folding, protein-membrane, and intra-protein interactions using single-bead-per-residue coarse-grained (CG) protein force fields [1-3]. These studies have shown that a cutoff of 0.8 nm reliably captures the effective residue-residue interaction strength in such CG protein models, and that the folding rates are largely insensitive to cutoff values up to 1.0 nm [3]. Therefore, we adopted a cutoff of 0.8 nm in our study.”

1. Souza, P. C. T., et al. (2025). *GōMartini 3: From large conformational changes in proteins to environmental bias corrections*. **Nature Communications**, 16(1), 4051.
2. Fuglebakk, E., Reuter, N., & Hinsen, K. (2013). *Evaluation of protein elastic network models based on an analysis of collective motions*. **Journal of Chemical Theory and Computation**, 9(12), 5618–5628.
3. Zhou, H., & Zhou, Y. (2002). *Folding rate prediction using total contact distance*. **Biophysical Journal**, 82(1), 458–463.

Form minor comments:

Point 5. The marker size in the legend could be further increased (The scatter points can remain as is).

Response: We thank the reviewer for this helpful suggestion. We have increased the marker size further in the Fig. 2H to improve visibility, while keeping the scatter point sizes in the plots unchanged as recommended.

Point 9. The reframed sentence is more or less same as the previous version and does not address the issue. The contact numbers for the different residue types are not that different and are possibly within error bars of each other. This leads me to an additional comment: There are no error bars on any of the bar plots in Fig. 5. I believe that the error bars are essential to support/deny the authors claims that polar and charged residues are the primary drivers of phase separation in their study. I suspect that the error bars would indicate that the contact numbers are within error limits of each other.

Response: We thank the reviewer for pointing out the missing error bars in Figure 5. We have now added error bars to all bar plots in Figure 5 and Figure S13. The updated figures confirm that the polar and charged residues of Tau make a higher contribution to intermolecular interactions. These results support that polar and charged residues are the primary drivers of phase separation. The Methods section has been revised accordingly.

“To estimate the error bars for intermolecular and intramolecular contact numbers, we calculated the contact number for four equal segments of the final 1 μ s of the trajectory. The

error bars correspond to the minimum and maximum contact number values obtained from these four segments.”

Figure 5. Molecular interactions driving the phase separation of Tau in the absence and presence of A β 40.

A, Intra-molecular and inter-molecular contact numbers in the Tau system and Tau-A β 40 systems.

B, Inter-molecular contact numbers between each pair of Tau residues in the Tau system. The accumulated contribution of each Tau residue to inter-molecular contacts is shown on the right.

C, Accumulated contact numbers of each Tau residue type with all residues, i.e., polar (P), charged (C), hydrophobic (H), and other (O) residues, in the Tau alone system.

D, Inter-molecular contact numbers between each Tau residue and each Tau/A β 40 residue in the Tau-A β 40 system. The accumulated contribution of each Tau residue to inter-molecular contacts is shown on the right.

E, Accumulated contact numbers of each pair of residue types in the Tau-A β 40 system. Contributions of each Tau residue type in Tau-Tau interaction, each Tau and A β 40 residue type in Tau-A β 40 interaction.

Figure S13. Accumulated inter-A β 40 contact numbers of each pair of A β 40 residue types in the Tau-A β 40 system.

Reviewer #1 (Remarks to the Author):

I would like to thank the authors for addressing my comments and answering my previous questions. After going through the responses, I still have some concerns about the contact analysis that has been performed:

Author reply: We sincerely thank the reviewer for the careful reading of our manuscript/response letter and for the thoughtful, detailed comments. We have addressed the concerns and have recalculated the inter-residue contacts using the residue-specific cutoff schemes suggested by the reviewer. The revised analyses lead to a more solid conclusion that charged residues are the primary drivers of Tau phase separation. Due to the change in cutoff scheme, the three citations related to the global cutoff of 0.8 nm have been removed in the revised manuscript.

Point 7. The primary concern about using a universal cutoff distance to determine contacts still holds. The three specific references that the authors have mentioned are not appropriate for the simulations/analysis performed in this study.

(a) Souza, P. C. T., et al. (2025). GōMartini 3: From large conformational changes in proteins to environmental bias corrections. *Nature Communications*, 16(1), 4051. This paper involves GoMartini simulations which are different from the CG HPS simulations, not just in terms of the resolution but also the underlying concept. In Go simulations, there is a template ‘native’ structure and the ‘contacts’ are defined relative to that structure. This is different from the contacts in the current study.

(b) Fuglebakk, E., Reuter, N., & Hinsen, K. (2013). Evaluation of protein elastic network models based on an analysis of collective motions. *Journal of Chemical Theory and Computation*, 9(12), 5618–5628. This paper uses elastic network models which are again different from the CG HPS simulations performed in the current study. Also, from a cursory look, the 0.8 nm cutoff that they mention is regarding the bonds in the model, not residue-residue contacts.

(c) Zhou, H., & Zhou, Y. (2002). Folding rate prediction using total contact distance. *Biophysical Journal*, 82(1), 458–463. This paper talks about “total contact distance” which is a combination of “contact order” and “long range order”. This has nothing to do with the coarse-grained HPS simulations that have been performed in the current study.

Author reply: We agree with the reviewer that the simulation setups and force fields in the cited references differ from those used in our work. In the original manuscript, our intention in citing these studies was not to suggest that GoMartini, elastic network models, or contact-order analyses are directly comparable to the coarse-grained HPS framework. Rather, we aimed to illustrate that a cutoff distance of 0.8 nm is a commonly used threshold for identifying noncovalent residue-residue contacts in protein systems.

In the first reference, a “native contact” was defined when two backbone beads (separated by at least two residues) lies within a cutoff distance of 0.8 nm, and a harmonic elastic bond was added between them to maintain native structures. Similarly, in the second reference, the 0.8 nm threshold was employed to identify residue pairs forming “native contact” when constructing elastic network

bonds. In both studies, these “bonds” are not chemical bonds but soft constraints introduced to maintain native structures, and the 0.8 nm cutoff is used as a criterion of a native noncovalent interaction formation.

In the third reference, the calculation of total contact distance (defined to predict protein folding rates) requires a residue-contact cutoff (R_{cut}) and a residue separation cutoff (l_{cut}). By comparing prediction results obtained under a wide range of R_{cut} values with experimental data, it was found that the results were largely insensitive to the residue-contact cutoff selected. This insensitivity lends further support to the suitability of using a cutoff near 0.8 nm as a reasonable indicator of residue-residue contact.

In my opinion, the references stated are inaccurate given the coarse-grained simulations that have been performed and therefore the reasoning for the 0.8 nm cutoff does not hold true. Instead, the authors should use residue specific cut-offs as is the norm. As an example, please refer to this paper: “Feito, Alejandro, et al. "Capturing single-molecule properties does not ensure accurate prediction of biomolecular phase diagrams." *Molecular Physics* 122.21-22 (2024): e2425757.” See Supplementary material section SVI, where a cut-off of 1.2ij has been used. Having said that, it would also be my recommendation to the authors to go through the references once more to make sure that they are accurate.

Author reply: We sincerely thank the reviewer for this insightful suggestion and for pointing us to the relevant reference. By searching the literature, we found that both global cutoff schemes and residue-specific cutoff schemes have been employed to determine residue-residue contacts/interactions in HPS simulations. Some examples are given below.

For the **global cutoff approaches**, Kang et al. performed CG HPS simulations to investigate the phase behavior of random peptide chains containing arginine and isoleucine, and defined a contact between two chains when the distance between any pair of beads was less than 0.68 nm^[1]. Zippo et al. used a one-bead-per-residue HPS model to study the phase separation of wild-type and phosphorylated TDP-43 and employed a global cutoff distance of 1.0 nm to determine inter-residue contacts/interactions^[2]. For the **residue-specific cutoff approaches**, in the study of RNA-mediated phase separation of FUS protein, Tejedor et al. defined interactions using a residue-specific cutoff of 1.2 σ ^[3]. Likewise, in the *Molecular Physics* (122.21-22 (2024): e2425757) paper, a residue-specific cutoff distance of 1.2 σ was also used to characterize residue-residue interactions in hnRNPA1^[4]. **The residue-specific cutoff scheme enables precise measurement of residue-residue interactions by accounting for the size of each individual residue. In contrast, the global cutoff scheme offers a rough but easy-to-implement estimation of such interactions by using a universal cutoff value.**

Following this reviewer’s suggestion, we have revised our calculations of intra- and intermolecular contact numbers using a residue-specific cutoff of 1.2 σ . While the overall interaction patterns in the Tau-alone and Tau-A β 40 systems remain qualitatively unchanged regardless of the cutoff scheme, the revised calculations highlight the dominant contributions of charged residues in Tau-involved interactions, whose contact numbers are significantly higher than those of other residue types,

demonstrating the leading role of electrostatic interactions in Tau and Tau-A β 40 phase separation. The importance of electrostatic interaction is also clearly seen in residue-residue contact maps (Figure 5B and 5D). We have revised the manuscript accordingly.

For Tau-alone system: “To further dissect these contributions, we calculated the accumulated contact numbers of different residue types. Our calculations indicated that charged residues were the primary drivers of Tau phase separation, with hydrophobic and polar residues also making nonnegligible contributions (Fig 5C).”

For Tau-A β 40 system: “High contact numbers were observed for hydrophobic-residue-rich regions of A β 40, i.e., residues 15-22 (₁₅QKLVEFAE₂₂) and residues 30-40 (₃₀AIIGLMVGGVV₄₀) (Fig 5D & S12). Analysis of accumulated contact numbers for each residue type showed that charged residues of Tau protein dominate Tau-A β 40 interactions followed by polar and hydrophobic residues, while hydrophobic residues of A β 40 dominate the Tau-A β 40 interaction followed by charged residues (Fig 5E). In addition, intermolecular interactions between A β 40 molecules were relatively weak (Fig S13), suggesting a minimal contribution to phase separation. Together, these results demonstrate that A β 40 interacts with Tau mainly through an interplay of hydrophobic and electrostatic interactions.”

For the summary: “Collectively, our simulations revealed that, A β 40 engages with Tau through its charged and hydrophobic residues, thereby promoting Tau-Tau intermolecular electrostatic interactions, and, to a lesser extent, hydrophobic and polar interactions. These combined effects ultimately enhance the phase separation of Tau.”

The Method section: “The cutoff for two residues forming a contact is calculated by $1.2 * (\sigma_i + \sigma_j) / 2$, in which σ_i and σ_j refer to the effective particle size defined as the distance at which the LJ potential equals zero^[3-4]. The number of contacts between two protein molecules was defined as the number of intermolecular bead pairs within this cutoff distance.”

Figure 5. Molecular interactions driving the phase separation of Tau in the absence and presence of Aβ40.

A, Intra-molecular and inter-molecular contact numbers in the Tau system and Tau-Aβ40 systems.

B, Inter-molecular contact numbers between each pair of Tau residues in the Tau system. The accumulated contribution of each Tau residue to inter-molecular contacts is shown on the right.

C, Accumulated contact numbers of each Tau residue type with all residues, i.e., charged (C), hydrophobic (H), polar (P), and other (O) residues, in the Tau alone system.

D, Inter-molecular contact numbers between each Tau residue and each Tau/Aβ40 residue in the Tau-Aβ40 system. The accumulated contribution of each Tau residue to inter-molecular contacts is shown on the right.

E, Accumulated contact numbers of each pair of residue types in the Tau-Aβ40 system. Contributions of each Tau residue type in Tau-Tau interaction, each Tau and Aβ40 residue type in Tau-Aβ40 interaction.

Figure S13. Accumulated inter-A β 40 contact numbers of each pair of A β 40 residue types in the Tau-A β 40 system.

References:

1. Kang W B, Bao L, Zhang K, et al. Multi-scale molecular simulation of random peptide phase separation and its extended-to-compact structure transition driven by hydrophobic interactions[J]. *Soft Matter*, 2023, 19(41): 7944-7954.
2. Zippo E, Dormann D, Speck T, et al. Molecular simulations of enzymatic phosphorylation of disordered proteins and their condensates[J]. *Nature Communications*, 2025, 16(1): 4649.
3. Tejedor A R, Sanchez-Burgos I, Estevez-Espinosa M, et al. Protein structural transitions critically transform the network connectivity and viscoelasticity of RNA-binding protein condensates but RNA can prevent it[J]. *Nature communications*, 2022, 13(1): 5717.
4. Feito A, Sanchez-Burgos I, Rey A, et al. Capturing single-molecule properties does not ensure accurate prediction of biomolecular phase diagrams[J]. *Molecular Physics*, 2024, 122(21-22): e2425757.

Point 9.

Fig. 5C, x-labels are missing. Since the contact analysis was erroneous, we cannot make any comment on the contact maps and therefore the subsequent analysis presented in Fig. 5.

Author reply: The missing x-labels in Fig. 5C have now been added. In addition, the contact analysis has been revised using the residue-specific cutoff scheme, as detailed in our response to the previous question.

Reviewer #3 (Remarks to the Author):

NCOMMS-25-56368: Alzheimer's A β 40 Catalyzes Tau Phase Separation and Aggregation via Early Nanocluster Solubilization

In this manuscript the authors have investigated the interplay of A β 40 peptides with Tau protein that drives the liquid-solid transitions of Tau protein droplets. The authors have employed various experimental techniques such as fluorescence microscopy, FRAP, turbidity measurements, NMR to qualify and quantify Tau aggregation phenotype in the presence of A β 40 peptides, and performed coarse-grained molecular dynamics (MD) simulations to understand the protein interactions at the residue level.

Key results: The authors claim that A β 40 promotes Tau phase separation and accelerates its phase transition to form amyloid like solid aggregates. Through MD simulations, they showed that A β 40 and Tau interact mainly through electrostatic and hydrophobic interactions. They also claim that in the absence of crowders, addition of A β 40 reduced cell toxicity, while in the presence of crowders (PEG), A β 40 increased cell toxicity. Based on the observations, the authors have proposed a mechanism where A β 40 can interact with Tau to keep it in its monomeric form. However, in the presence of crowders, A β 40 also promotes the phase separation and transition of tau proteins, leading to disease pathology.

Overall, the manuscript is well-written and presents some interesting results. However, there are several conceptual and technical points, which call into question the certain conclusions of the work.

Major comments:

1. The phrase "A β 40 does not undergo LLPS" has little meaning from a thermodynamic standpoint. Since phase separation is a thermodynamic process, the conditions must be specified. Also the authors used the catGRANULE algorithm to predict whether A β 40 undergoes LLPS. conditions is this algorithm predicting the phase separation propensity for? Related to that, what does the score from this algorithm mean. Is there a maximum/minimum value one can expect? Is there a threshold score that makes something undergo LLPS (say under physiological conditions)? The details there are quite vague and it is difficult to appreciate what exactly is being predicted.
2. I find it difficult to see how the droplet area comparisons with and without A β 40 directly demonstrate accelerated kinetics of droplet formation. A β 40 alone produces droplets with a notably different topology than Tau alone (see Fig. 2A). Thus, the observed differences in droplet areas could largely reflect intrinsic differences in droplet morphology rather than kinetic effects. Consequently, additional evidence beyond droplet area measurements is required to substantiate the claim that A β 40 accelerates the kinetics of Tau droplet formation.
3. From the coarse-grained MD simulations that authors claim that the dense-phase density in the presence of A β 40 is greater than when A β 40 is absent. However, the phase diagram in Figure 4F show practically identical dense-phase densities for both systems. Therefore, their results do not support this claim. Furthermore, there are no error bars shown in the figure. What is the error associated with the critical point estimate? It is likely that the 2 K difference in T_c is within the uncertainty. Based on the results presented so far, it does not seem, from a thermodynamic standpoint, that the A β 40 is having a significant impact.
4. Lines 366-368: "When Tau was co-incubated with A β 40, the aggregates displayed slightly increased toxicity compared to Tau alone (Fig 6B)." The difference there does not seem significant. They are both within the errors.
5. Fig. 1F: At what time point of the droplet formations were the FRAP measurements performed? Something that seems contradictory to me is that the Tau droplet sizes increase rapidly in the presence of A β 40 due to coalescence (indicating liquid-like nature), while at the same time FRAP measurements are showing arrested dynamics. I would not have expected the two to be true at

the same time. Were the FRAP measurements performed long after (example, a day) the large droplets were already formed? Also, are similar FRAP measurements performed for A β 40 to confirm that all the dynamics are arrested?

6. Fig. S9, Line 309 – 313: (1) In the MSD calculation for the Tau+A β 40, were the proteins treated separately? Tau is significantly longer than A β 40, which could lead to varying diffusion coefficient for the two species. It might be useful to separate the two to be accurate and maybe get some useful insight. (2) How exactly were the diffusion coefficient measured? Did the authors assume diffusive behavior and use the Stokes-Einstein relation? If so, that might be wrong since the MSD curves are clearly not linear and indicate sub-diffusive behavior (as expected). In that case one should be able to fit the MSD curve to $\langle r^2(\tau) \rangle = K_\alpha \tau^\alpha$. Overall, the details on how the MSD calculations were done are lacking. What ensemble was used, how long were the simulations etc?
7. Line 317-318: Provide more information as to how the interaction analysis was performed. Was a distance threshold used? If so, what was it? Also, were the contact maps normalized in Fig. 5? Between residues 200-300 the contact maps have some contact values almost equal to 1. Does this mean that those residue pairs were in contact most of the time?

Minor comments:

1. Figure 6 B/E or C/F: Invert either the intensity or the cell viability so that both quantities vary in the same direction. This will make the comparison of B and C and E and F more straightforward.
2. Lines 155-156: Which figures correspond to the brightfield microscopy images that is being alluded to?
3. Fig. 1A: Indicate the sequence of the Tau protein that has been used (and the uniprot id). Is it P10636-8?
4. Line 169-170: There is a typo. Instead of “with or with” it should be “with or without”
5. Fig. 2H: The legend markers should be made larger.
6. Line 249: Sentence reads odd. Missing word here “formed relatively nanocluster”?
7. Line 274: Typo. Instead of “imply that enhancement A β 40–Tau interactions”, it should be “imply enhancement of A β 40–Tau interactions”
8. Fig. 5C: It would be helpful if you could indicate in the caption what the letters ‘H’, ‘P’, ‘C’, ‘O’ represent.
9. Line 333 – 335: “*Our calculations indicated that charged residues were the primary drivers of Tau phase separation, followed by polar residues, with hydrophobic residues also making non-negligible contributions (Fig 5C).*” The way this sentence is written, it seems that the accumulated contact numbers for the electrostatic interactions are much larger than the other ones (polar and hydrophobic). However, from the figure the polar contacts are almost equal to the electrostatic ones and closely followed by the hydrophobic interactions. The sentence could perhaps be framed in a different manner.